# Introducing Rhetorical Parallelism Detection: A New Task with Datasets, Metrics, and Baselines

**Stephen Bothwell**
University of Notre Dame
sbothwel@nd.edu

**Justin DeBenedetto**
Villanova University
justin.debenedetto@villanova.edu

**Theresa Crnkovich**   **Hildegund Müller**   **David Chiang**
University of Notre Dame
{tcrnkovi,hmuller,dchiang}@nd.edu

## Abstract

Rhetoric, both spoken and written, involves not only content but also style. One common stylistic tool is *parallelism*: the juxtaposition of phrases which have the same sequence of linguistic (*e.g.*, phonological, syntactic, semantic) features. Despite the ubiquity of parallelism, the field of natural language processing has seldom investigated it, missing a chance to better understand the nature of the structure, meaning, and intent that humans convey. To address this, we introduce the task of *rhetorical parallelism detection*. We construct a formal definition of it; we provide one new Latin dataset and one adapted Chinese dataset for it; we establish a family of metrics to evaluate performance on it; and, lastly, we create baseline systems and novel sequence labeling schemes to capture it. On our strictest metric, we attain $F_1$ scores of 0.40 and 0.43 on our Latin and Chinese datasets, respectively.

## 1 Introduction

*Ueni, uidi, uici*,[1] or, "I came, I saw, I conquered" – why is this saying so memorable? One reason is the high degree of *parallelism* it exhibits in its morphology (each verb is first-person, singular, perfect tense), phonology (each verb starts with /w/ and ends with /iː/), and prosody (each verb has two long syllables with accent on the first). In similar sayings (as in Fig. 1), syntax and semantics also contribute. The related elements generally occur in either the same order or in an inverted order.

Parallelism is organically employed in many argumentative structures (*e.g.*, "on the one hand . . . on the other hand . . ."), making parallelism a routine rhetorical figure. The rhetorical impact of

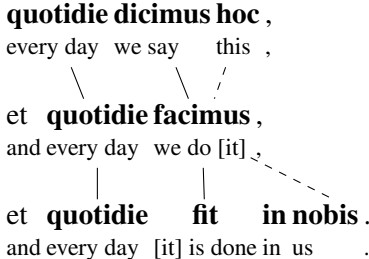

Figure 1: Example three-way parallelism from Augustine's *Sermones* (Sermon 181, Section 8). Solid lines connect words with multiple linguistic features in common; dashed lines indicate a weaker connection.

these structures is apparent even for audiences not schooled in classical rhetoric. Recognition of parallelism, then, is important for grasping the structure, meaning, and intent that humans wish to convey; thus, the computational modeling of parallelism is both an interesting and practical problem.

However, because parallelism can occur at so many levels linguistically—often with no lexical overlap—modeling parallelism computationally is difficult. As a first foray into studying the problem of rhetorical parallelism detection (RPD) and enabling others to study it, we present in this paper multiple public datasets, metrics, and models for it.

For one dataset, we turn to the Latin sermons of Augustine of Hippo (354–430). Augustine had been trained as a rhetorician and had taught the craft of secular rhetoric before his conversion to Christianity. However, upon becoming a preacher, he consciously replaced the adorned style of late ancient rhetoric with a style streamlined toward speaking well with diverse North African congregations. Parallelism frequently marked this style.

Augustine did not just employ parallelism stylistically, however; he also theorized about it. In his work *De Doctrina Christiana*, Augustine described three rhetorical styles, or *genera dicendi* ("ways of

---

[1] Popularly attributed to Julius Caesar (Suetonius Tranquillus, 1993). In Classical Latin, *u* and *v* were graphical variants of one letter; as a consonant, it was pronounced /w/. Following common practice, we write it as *u*. We also write *i* and *j* as *i*.

speaking") (Augustine of Hippo, 1995); from these, he characterized the *genus temperatum* ("moderate style") by highly-parallel passages.[2] Thus, his sermons are ideal for studying parallelism. In addition, his theory implies that parallelism detection may be useful for automatic stylistic analysis. Already, it has been used to analyze discourse structure (Guégan and Hernandez, 2006), summarize documents (Alliheedi, 2012), identify idioms (Adewumi et al., 2022), evaluate student writing (Song et al., 2016), study political speech (Tan et al., 2018) and detect fake news (Gautam and Jerripothula, 2020).

Parallelism detection may also have broader NLP applications. In syntactic parsing, in the sentence *I saw a man with a mustache and a woman with a telescope*, the reading where *telescope* modifies *saw* is all but ruled out because it would violate parallelism. Thus, modeling parallelism could assist in syntactic disambiguation. Parallelism is also vital in disfluency detection, as speakers tend to maintain prior syntactic context when amending verbal errors (Zayats and Ostendorf, 2019).

Toward such ends, we establish the task of rhetorical parallelism detection (RPD, Section 2). We create one dataset from Augustine's sermons and adapt another consisting of Chinese student essays (Section 4), establish evaluation metrics (Section 5), and investigate baseline models (Section 6). By automatically learning and exploiting relationships among linguistic features, we achieve roughly a 40% $F_1$ score on both datasets' test sets (Section 7).

## 2   Task Definition

Moving away from the sentence-level conceptions of Guégan and Hernandez (2006) and Song et al. (2016), we formalize the task of **rhetorical parallelism detection** (RPD). Broadly, we deem sequences to be *parallel* if they meet two conditions:

1. **Locality**: Parallel structures should be within temporal proximity of one another for two related reasons. First, structures that are close by are more likely to be intentional rather than incidental in the mind of the speaker/author. Second, for parallel structures to be rhetorically effective, they must be consecutive enough to be recalled by the listener/reader.

2. **Linguistic Correspondence**: Parallel structures should have some linguistic features in common, which could include features of

phonology, morphology, syntax, semantics, or even *style*. The features could be the same (*e.g.*, the repeated sounds in *ueni, uidi, uici*) or diametrically opposed (*e.g.*, the opposite meanings in *a time to be born and a time to die*). These features typically occur in the same order or the opposite order.

Suppose that we have a document **w**, which is a sequence of tokens $w_1 \ldots w_n$. A **span** of **w** is a pair $(i, j)$, where $1 \le i \le j \le n$, whose **contents** are the tokens $w_i \ldots w_j$. We say that two spans $(i_1, j_1)$ and $(i_2, j_2)$ are **overlapping** if they have at least one token in common—that is, $i_1 \le i_2 \le j_1$ or $i_2 \le i_1 \le j_2$. Meanwhile, they are **nested** if $i_1 \le i_2 \le j_2 \le j_1$ or $i_2 \le i_1 \le j_1 \le j_2$.

A **parallelism** is a set of two or more non-overlapping spans whose contents are parallel. We call these spans the **branches** of a parallelism. An example of a complete parallelism with three branches is given in Fig. 1. RPD, then, is the task of identifying all of the parallelisms in a text.

Let *P* be a set of parallelisms. Then *P* falls into one of three categories:

1. *P* is **flat** if all the branches of every parallelism in *P* are pairwise non-overlapping.

2. *P* is **nested** if some pair of its branches nest and no pair of its branches overlap without also being nested.

3. *P* is **overlapping** if it contains some pair of branches which overlap.

Our Augustinian Sermon Parallelism (ASP) dataset (see Section 4) is nested, and the Paibi Student Essay (PSE) dataset (Song et al., 2016) is flat.[3] Although our baseline models only predict flat sets of parallelisms, our evaluations include nested ones.

## 3   Related Work

In this section, we connect two subareas of NLP to RPD in order to display their relation to and influence on its development.

### 3.1   Automated writing evaluation

Arguably, the main research area that has explored RPD is automated writing evaluation (AWE). Since its inception (Page, 1966), AWE has aimed to reduce instructor burden by swiftly scoring essays;

---

[2]See chapters 4.20.40; 20.44; 21.47f.

[3]Song et al.'s dataset was not originally named but has been given a name through its inclusion in this work.

| Dataset | Documents | Sections | Tokens | Branched Tokens | Branches | Parallelisms |
|---|---|---|---|---|---|---|
| ASP | 80 | 477 | 134,956 | 19,701 | 4,651 (39) | 2,062 (14) |
| PSE-I | 409 | 3,855 | 241,203 | 25,529 | 2,153 (0) | 786 (0) |

Table 1: Dataset frequency statistics for both of the datasets featured in this paper. "Sections" are defined as natural partitions of documents, such as divisions determined by editors (ASP) or natural paragraphs (PSE-I). "Branched Tokens" refer to all tokens involved in a branch. Parenthesized values refer to the nested subset of a given quantity.

| Dataset | Parallelisms per Section | Branches per Parallelism | Branch Distance (Tokens) | Branch Size (Tokens) | NLO (Branches) | % Pairs with No LO |
|---|---|---|---|---|---|---|
| ASP | 4.32 ± 3.22 | 2.26 ± 0.68 | 2.54 ± 2.29 | 4.24 ± 2.72 | 0.24 ± 0.19 | 24.17% |
| PSE-I | 1.92 ± 1.47 | 2.74 ± 0.75 | 2.41 ± 4.00 | 11.86 ± 7.09 | 0.31 ± 0.15 | 2.30% |

Table 2: Dataset derived statistics. All figures but the last one are means with standard deviations. The lexical overlap (LO) metrics are normalized with respect to the union of branches' token multisets.

however, especially with the involvement of neural networks, it seems to be limited in its pedagogical benefit because of its inability to give sufficient feedback (Ke and Ng, 2019; Beigman Klebanov and Madnani, 2020).

Concerning RPD, Song *et al.* explored parallel structure as a critical rhetorical device for evaluating writing quality (Song et al., 2016). Construing parallelism detection as a sentence pair classification problem, they achieved 72% $F_1$ for getting entire parallelisms correct on their dataset built from Chinese mock exam essays via a random forest classifier with hand-engineered features. Subsequent work used RNNs (Dai et al., 2018) and CNNs with some custom features (Liu et al., 2022). Such work has also been applied in the IFlyEA assessment system to provide students with stylistic feedback (Gong et al., 2021).

Compared to that work, our formalization of RPD permits a token-level (as opposed to sentence-level) granularity for parallel structure. In line with this, we approach RPD in terms of sequence labeling instead of classification. Moreover, we provide the first (to our knowledge) public release of token-level parallelism detection data (see Section 4).

## 3.2 Disfluency detection

RPD closely resembles and is notably inspired by disfluency detection (DD). DD's objective is generally to detect three components of a speech error: the *reparandum* (the speech to be replaced), the *interregnum* (optional intervening speech), and the *repair* (optional replacing speech) (Shriberg, 1994). Because the *reparandum* and *repair* often relate in their syntactic and semantic roles, they (as with parallelisms) share many linguistic features.

DD has frequently been framed as a sequence labeling task. Some previous models use schemes which adapt BIO tagging (Georgila, 2009; Ostendorf and Hahn, 2013; Zayats et al., 2014, 2016), while others (Hough and Schlangen, 2015) have proposed novel tagging schemes on the basis of prior theory (Shriberg, 1994) and the Switchboard corpus's notation (Meteer and Taylor, 1995), augmenting tags with numerical links to indicate relationships between reparanda and repairs.

We employ elements of both types of tagging schemes: we directly adapt BIO tags to parallelisms, and, like Hough and Schlangen (2015), we also augment tags with numerical links to indicate relationships between branches (see Section 6.1).

## 4 Datasets

This paper presents two datasets for RPD: the Augustinian Sermon Parallelism (ASP) and Paibi Student Essay (PSE) datasets. We first describe steps taken for the individual datasets in Sections 4.1 and 4.2 before discussing shared preprocessing steps and data analyses in Sections 4.3 and 4.4.

### 4.1 Augustinian Sermon Parallelism Dataset

The ASP dataset consists of 80 sermons from the corpus of Augustine of Hippo (Augustine of Hippo, 2000a,b).[4] Our fourth author, an expert classicist and Augustine scholar, labeled these sermons for parallel structure using our annotation scheme. This scheme involves labeling branches and linking them to form parallelisms. We further distinguish between synchystic (same order) and chiastic (in-

---

[4]The ASP dataset is freely available at https://github.com/Mythologos/Augustinian-Sermon-Parallelisms.

(a) *ut     ipse*  [ *panis        esuriret*              ]₁ , [ *satietas      sitiret*                ]₁ , [ *uirtus*
so.that itself  bread:NOM;SG hunger:IPFV;SBJV;3SG  ,  satiety:NOM;SG thirst:IPFV;SBJV;3SG  ,  strength:NOM;SG

*infirmaretur*            ]₁ , [ *sanitas        uulneraretur*          ]₁ , [ *uita      moraretur*          ]₁ ?
weaken:IPFV;SBJV;3SG  ,  health:NOM;SG wound:IPFV;SBJV;3SG  ,  life:NOM;SG die:IPFV;SBJV;3SG  ?

"... so that bread itself might hunger, satiety might thirst, strength might be weakened, health might be wounded, life might die?"

(b) [ 小草   更   绿   ]₁ , [ 天空   更   蓝   ]₁ , [ 生活   更   美好   ]₁ .
xiǎocǎo gèng lǜ   ,  tiānkōng gèng lán  ,  shēnghuó gèng měihǎo  .
grass   more green  ,  sky    more blue  ,  life     more good   .

"The grass is greener, the sky is bluer, and life is better."

Figure 2: Examples from the ASP and PSE datasets. Blue square brackets demarcate each branch, and numbering indicates the parallelism to which each branch belongs. We provide a word-by-word English translation, gloss some morphological features in branches, and present an idiomatic translation. Example (a) is from Sermon 207, Section 1. Example (b) is from Essay 1, Paragraph 5. For more ASP dataset examples, see Figs. 7 and 8.

verted order) parallelisms.[5] For more details on our annotation scheme, see Appendix A; for details on a verification of the dataset's quality through an inter-annotator agreement study, see Appendix C.

An example of Augustine's use of parallelism is presented in Fig. 2(a). In this subordinate clause, Augustine builds up a five-way parallelism. This parallelism not only boasts shared morphology and syntax through a series of subject-verb clauses, but it also presents *stylistic* parallelism. Each clause uses the rhetorical device of personification, presenting an object or abstract idea as an agent, and this device is paralleled to emphasize it. As this example shows, Augustine frequently layers linguistic features to compose parallelisms.

To preprocess the ASP data, we remove all capitalization, following previous work on Latin word embeddings (Bamman and Burns, 2020; Burns et al., 2021). We also employ the LatinWordTokenizer from the CLTK library to tokenize our data (Johnson et al., 2021).

## 4.2 Paibi Student Essay Dataset

The PSE dataset was created by Song et al. (2016) to improve automatic essay evaluation.[6] They collected a set of mock exam essays from Chinese senior high school students. Two annotators then marked parallel structure in these essays. Annotators labeled parallelism on the *sentence* rather than *span* level. On this task, they achieved an inter-annotator agreement of 0.71 according to the $\kappa$ statistic (Carletta, 1996) over a set of 30 essays.

In Chinese, a 排比 (*pǎibǐ*, parallelism) is some-

times defined as having at least three branches (although the PSE dataset has many examples of two-branch parallelisms). One such three-way parallelism is given in Fig. 2(b), where both lexical repetition of the comparative adverb 更 (*gèng*) and syntactic parallelism help the sentence's three clauses to convey its message in a spirited manner.

We obtained 409 of the original 544 essays from the authors. The essays are annotated for inter-paragraph parallelisms, intra-paragraph parallelisms, and intra-sentence parallelisms. To adapt them for our use, we first excluded inter-paragraph parallelisms, as these do not satisfy our criterion of locality. Then, for each intra-sentence parallelism that had been annotated, our annotators marked the locations of the parallel branches. For more details about this annotation process, see Appendix A. We call this version of the dataset PSE-I, or the version only having parallelisms *inside* paragraphs. We retained the tokenization generated by the Language Technology Platform (LTP) (Che et al., 2010).

## 4.3 Data collection and preprocessing

Both datasets had annotations collected for them via the BRAT annotation tool (Stenetorp et al., 2012). In applying the parallelism annotations to the data, both datasets excluded punctuation from parallelisms. This was done to reduce annotator noise brought about by the inconsistent membership of punctuation in parallelisms.

After preprocessing each dataset, we split both into training, validation, optimization, and test sets. To do so, we attempted to keep the ratio of tokens inside parallelisms to tokens outside parallelisms as even as possible across each set, assuring that the sets were distributed similarly. For more details on our data splitting, see Appendix E.

---

[5]The terms *synchystic* and *chiastic* are used by analogy with the traditional rhetorical terms *synchysis* and *chiasmus*.

[6]The PSE dataset and its PSE-I variant are freely available at https://github.com/Mythologos/Paibi-Student-Essays.

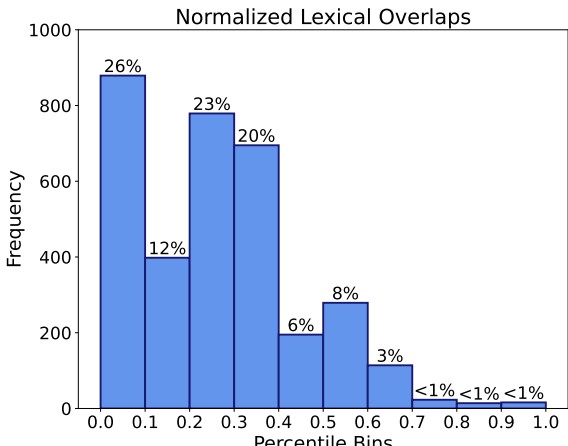

Figure 3: Histogram of normalized lexical overlap for all paired branches in the ASP dataset. Rounded percentages indicating the amount of data represented in each bin (of size 0.1) are placed above each bar.

## 4.4 Dataset statistics

To summarize our datasets, we provide a numerical and statistical overview in Tables 1 and 2.

One factor we wanted to examine was the type of similarity that parallel branches exhibit; are their similarities surface-level or more linguistically complex? To measure this, we compute the *normalized lexical overlap* (NLO) of all pairs of related branches. Treating each branch as a multiset of tokens, we apply the formula

$$\text{NLO}(b_1, b_2) = \frac{|b_1 \cap b_2|}{|b_1 \cup b_2|}$$

where $b_1, b_2$ are related branches. This value is 1 if $b_1 = b_2$, and it is 0 if $b_1 \cap b_2 = \emptyset$.

In Fig. 3, we depict a histogram of the ASP dataset's NLO across all related branch pairs. (The PSE-I dataset's histogram is similar, but it peaks between 0.2 and 0.3 and has few pairs between 0 and 0.1.) This histogram shows that the vast majority of related branch pairs are frequently lexically dissimilar from one another; most are below 0.6 NLO, showing that it is very common to have at least half the words in a parallelism be different from one another. Thus, as our task definition asserted, parallelisms exploit many linguistic features to produce an effect; in turn, any method for RPD should try to capture these relationships.

## 5 Evaluation Metrics

Next, we describe a general family of evaluation metrics for RPD and specify one instance of it.

Suppose that we have a document **w**, as above, and a reference set **G** and a hypothesis set **H** of parallelisms in **w**. An evaluation metric for RPD should check both that the branches in **H** have the same spans as those in **G** and that the branches are grouped into parallelisms in **H** as they are in **G**.

Given these criteria, we follow prior work in coreference resolution (CR) to establish our metrics. Coreference resolution involves detecting an entity of interest and linking all mentions related to that entity together; in a similar manner, our task links a set of spans together. As in the Constrained Entity-Alignment F-Measure metrics (Luo, 2005), we express our metrics as a bipartite matching between **G** and **H**. We use the Kuhn-Munkres (also known as "Hungarian") algorithm (Kuhn, 1955; Munkres, 1957) to maximally align parallelisms between **G** and **H**. To do this, we must define two functions: score and size.

The function $\text{score}(p_1, p_2)$, where $p_1$ and $p_2$ are parallelisms, determines how well $p_1$ matches $p_2$, and the function $\text{size}(p)$, where $p$ is a parallelism, bounds the score that $p$ can merit with another parallelism. These functions must satisfy $\text{size}(p) > 0$ and $0 \leq \text{score}(p_1, p_2) \leq \min\{\text{size}(p_1), \text{size}(p_2)\}$.

With those in place, we can find the bipartite matching with maximum weight $m$,

$$m = \max_M \sum_{(p_1, p_2) \in M} \text{score}(p_1, p_2)$$

where the maximization is over matchings $M$ between **G** and **H**. Having $m$, we define metrics in the likeness of precision ($P$), recall ($R$), and the $F_1$-score ($F_1$) as follows:

$$P = \frac{m}{\sum_{p \in \mathbf{H}} \text{size}(p)}$$
$$R = \frac{m}{\sum_{p \in \mathbf{G}} \text{size}(p)}$$
$$F_1 = \frac{2}{1/P + 1/R}.$$

The simplest choice of size and score is

$$\text{size}(p) = 1$$
$$\text{score}(p_1, p_2) = \begin{cases} 1 & \text{if } p_1 = p_2 \\ 0 & \text{if } p_1 \neq p_2. \end{cases}$$

We call this the *exact parallelism match* (EPM) metric, where each parallelism in **H** must perfectly match one in **G** to gain credit.

As an example of this metric, consider the passage given in Fig. 4. Suppose that the depicted parallelism—call it $p_1$—is the *only* parallelism in **G**. Furthermore, suppose that the model proposes two hypotheses, $\mathbf{H} = \{p_1, p_2\}$, where $p_2$ is a parallelism that does not overlap with $p_1$. Then

$$\text{score}(p_1, p_1) = 1 \qquad \text{score}(p_1, p_2) = 0$$

From this, we derive the maximally-weighted matching, $M = \{(p_1, p_1)\}$, and $m = 1$. Then

$$P = \frac{1}{2} \qquad R = \frac{1}{1} \qquad F_1 = \frac{2}{3}.$$

Appendix B introduces some alternative choices of `score` and `size` that give partial credit for imperfectly-matching hypothesized parallelisms.[7]

## 6 Models

In this section, we propose some models for automatic RPD as a starting point for future research. Our models here treat RPD as a sequence labeling problem, using several novel variants of BIO tagging (Ramshaw and Marcus, 1995).[8]

### 6.1 Tagging schemes

Branches are substrings, like named entities in NER, and parallelisms are sets of related substrings, like disfluencies in DD. Variants of BIO tagging have been successful for these sequence labeling tasks (Zayats et al., 2016; Reimers and Gurevych, 2017), so they are a natural choice for RPD as well. In this scheme, the first word of a branch is tagged B; the other words of a branch are tagged I; words outside of branches are tagged O.

However, BIO tagging does not indicate which branches are parallel with which. In each parallelism, for each branch $b$ except the first, we augment the B tags with a *link*, which is a number that indicates the previous branch with which $b$ is parallel. We propose two linking schemes.

Suppose we have consecutive parallel branches $b_1 = (i_1, j_1)$ and $b_2 = (i_2, j_2)$ with $j_1 < i_2$. A *token distance* link, akin to links used in DD (Hough and Schlangen, 2015), is $j_1 - i_2$; it is the (negative)

---

[7]We accompany this paper with the `pyrallelism` library. It implements each of our metrics and provides sample formats for evaluating RPD. It is available on PyPI and at `https://github.com/Mythologos/pyrallelism`.

[8]Our main modeling and results repository is available at `https://github.com/Mythologos/Intro-RPD`. Our models are implemented in PyTorch (Paszke et al., 2019) and are initially based on code by Robert Guthrie (2017).

number of word-to-word hops to get from $b_2$'s start to $b_1$'s end. A *branch distance* link is the (negative) number of branch-to-branch hops to get from $b_2$ to $b_1$. If $P$ contains the interlocking parallelisms $p_1 = \{(1, 3), (7, 9)\}$ and $p_2 = \{(4, 6), (10, 12)\}$, then the token distance between $(1, 3)$ and $(7, 9)$ is $-4$, while the branch distance is $-2$.

It is important to form tag sequences that help the model to learn better, diverging decisions from the dominant majority class O. Therefore, in addition to adding links, we include three additional tag types, each of which is exhibited in Fig. 4.

First, the M tag replaces O tags that occur for tokens that occur in the *middle* of consecutive branches in a parallelism. Lines 2 and 3 of Fig. 4 exhibit this: four non-branch tags become M instead of O because they are between related branches. This shift obstructs a model from predicting single-branch parallelisms by providing a separate pathway to seek linked branches, as O → B$\ell$ (where $\ell$ stands for a link) can never occur in the data.

Second, the E tag replaces branch-ending I tags to indicate the *end* of a branch. Adding this tag removes the I → O transition. This change may encourage a model to be more sensitive to a branch's endpoint; because most parallelisms are more than two tokens long, branches largely must transition from B to either I or E, and E must eventually be selected before returning to O.

Third, the J tag replaces I tags in non-initial branches, where an initial branch is the first branch of a parallelism that occurs in a document. When paired with M tags, J tags help to promote a sequence of transitions which do not include O as a likely candidate until two branches have concluded. While the second example of Figure 4 displays this, Figure 5 shows this behavior across ASP's entire training set. Treating a sequence of tags as a linear chain, we tally all node transition pairs. When compared to the BIO chain of tags (left), we can see that I → O no longer exists in the BIOMJ chain (right). Only non-initial branch tokens (*i.e.*, B$\lambda$ or J) can transition to O. Through its supported transitions, the altered tagset actively enforces the constraint that a parallelism must have at least two branches.

With these three new tags, we form eight tagsets—BIO, BIOE, BIOJ, BIOM, BIOJE, BIOME, BIOMJ, and BIOMJE—and apply each link type to them, forming sixteen tagging schemes. These tagging schemes constitute all parallelisms in each dataset: each level of nesting is an addi-

| | quotidie | dicimus | hoc | , | et | quotidie | facimus | , | et | quotidie | fit | | in | nobis | . |
|---|---|---|---|---|---|---|---|---|---|---|---|---|---|---|---|
| Latin | **quotidie** | **dicimus** | **hoc** | , | et | **quotidie** | **facimus** | , | et | **quotidie** | **fit** | | **in** | **nobis** | . |
| English | Every day | we say | this | , | and | every day | we do [it] | , | and | every day | [it] is done | | in | us | . |
| BIO-Token | B | I | I | O | O | B–3 | I | O | O | B–3 | I | | I | I | O |
| BIOMJ-Token | B | I | I | M | M | B–3 | J | M | M | B–3 | J | | J | J | O |
| BIOME-Branch | B | I | E | M | M | B–1 | E | M | M | B–1 | I | | I | E | O |

Figure 4: The example of Fig. 1 from ASP annotated with various tagging schemes. For line 1, bolded words are part of the parallelism, and all contiguous units represent branches. For line 2, items in brackets indicate words not directly present (but implied) in the original text. For lines 3–5, "Token" and "Branch" refer to link types.

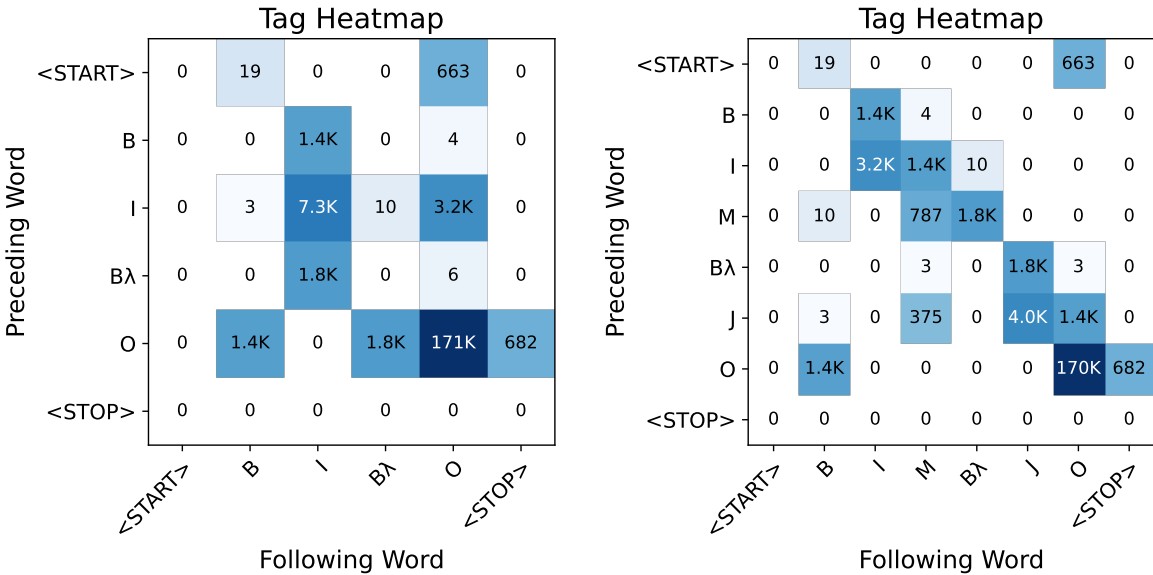

Figure 5: Two heatmaps tabulating tag transition frequencies across all (two) strata in ASP's training set. Darker colors indicate larger values; text colors change for readability. The <START> and <STOP> tokens refer to section beginnings and ends. The left and right heatmaps are for the BIO and BIOMJ tagsets, respectively.

tional layer, or *stratum* (pl. *strata*), of tags.

## 6.2 Architecture

Inspired by prior success in sequence labeling for named entity recognition, we employ a conditional random field (CRF) output layer (Lafferty et al., 2001) in combination with various neural networks (Huang et al., 2015; Ma and Hovy, 2016; Lample et al., 2016). Our general architecture (Fig. 6) proceeds in five steps. It *embeds* tokens (words or subwords) as vectors; it *blends* token-level embeddings into word-level embeddings; it *encodes* the sequence of embeddings to incorporate more contextual and task-specific information; it *maps* encodings to tag space with a linear layer; it uses a CRF to compute a distribution over tag sequences.

Each model examined in Section 7.2 is a variation on this paradigm. We tested a total of six models. Following work in NER, we selected a BiLSTM-CRF baseline (Huang et al., 2015; Ma and Hovy, 2016; Lample et al., 2016). We also tried exchanging the BiLSTM for a Transformer

(Vaswani et al., 2017) as an encoder layer.

We tried three embedding options. The first option was to learn embeddings from scratch. The second option was to use frozen word2vec embeddings (Mikolov et al., 2013). We selected a 300-dimensional embedding built from Latin data lemmatized by CLTK's LatinBackoffLemmatizer (Johnson et al., 2021) trained by Burns et al. (2021). The third option was to employ frozen embeddings from a BERT model—namely, Latin BERT (Bamman and Burns, 2020) for the ASP dataset and Chinese BERT with whole word masking (Cui et al., 2020, 2021) for the PSE-I dataset.

For both the Transformer encoder and the BERT embeddings, we applied WordPiece tokenization (Wu et al., 2016) by reusing the tokenizer previously trained for Latin BERT (Bamman and Burns, 2020) with the tensor2tensor library (Vaswani et al., 2018). We employed operations (termed *blending functions*) to combine subword representations into word representations. The choice of blending function did not heavily impact our

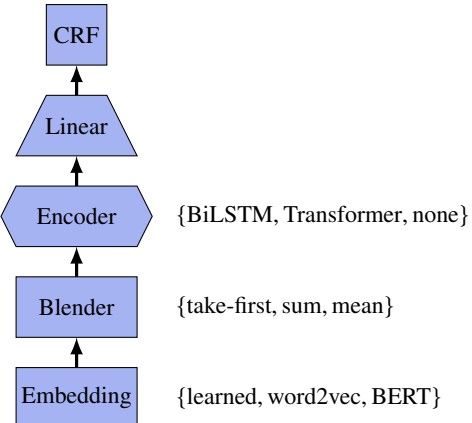

Figure 6: Generic Encoder-CRF architecture and per-level options examined. Shapes roughly indicate relative changes in representation sizes.

results, so we defer discussion of them to Appendix D.3. For other models, we did not use subwords, and no blending function was needed.

To reduce the average sequence length seen by our models, each sequence presented is a *section* rather than a whole document. For the ASP dataset, each sermon's division into sections was imposed by later editors of the texts. Meanwhile, for the PSE-I dataset, we split the data based on paragraph divisions. When sections remain longer than BERT's 512-token limit, we split sections into 512-token chunks and concatenate the output sequences after the embedding layer. As noted in previous CR work (Joshi et al., 2019), this approach is superior to merging overlapping chunks.

We discuss other design choices in Appendix D.

# 7 Experiments

In this section, we describe our experiments (Section 7.1 and present our results (Section 7.2).

## 7.1 Experimental design

For each dataset, we performed hyperparameter searches over several model architecture and tagging scheme combinations. We judged these as the elements which would likely primarily govern task performance. For the ASP dataset, we trained six possible architectures described in Section 6.2 with each of the sixteen tagging schemes described in Section 6.1. Meanwhile, for the PSE-I dataset, we ran the three BERT-based architectures with each of the same sixteen tagging schemes. The hyperparameters for each trial were chosen using random search (Bergstra and Bengio, 2012) described in Appendix F. Eight trials per configuration were

trained for the ASP and PSE-I datasets, totaling 768 and 384 experiments, respectively.

Each trial's model trained for up to 200 epochs using Adam (Kingma and Ba, 2015) and gradient clipping with an $L_2$ norm of 1 (Pascanu et al., 2013). We stopped training early if the model's $F_1$ score did not improve for 25 epochs on the validation set using the maximum branch-aware word overlap (MBAWO) metric (defined in Appendix B).[9] Each trial was assessed on the optimization set. We denoted the trial with the highest MBAWO score on this set per setting as that setting's best run. We evaluated each best run on the test set.

## 7.2 Results

To compare the performance across models, we determined the best result for each model, as shown in Table 3. "Best" is defined as the experiment with the highest $F_1$ score on the test set across all attempted settings. We also highlight a few factors which generally improved performance: the embeddings, encoders, and tagsets selected.[10]

In terms of *embeddings*, BERT embeddings vastly improved performance. As Table 5 depicts, BERT-based models exceeded every other non-BERT model by at least 0.2 $F_1$ on the ASP dataset. We attribute their superiority to the contextual representations produced by BERT. This is supported by the fact that both Burns et al.'s embeddings and Latin BERT's embeddings were mainly trained on the same Internet Archive dataset (Bamman and Crane, 2011; Bamman and Smith, 2012).

In terms of *tagging schemes*, we wanted to know whether any schemes performed significantly better than the others. We applied the Friedman ranked-sums test (Friedman, 1937, 1940). Although this test is usually used to compare multiple classifiers across multiple datasets (Demšar, 2006), we instead took results across a single dataset (ASP) and sampling procedure (our hyperparameter search process) and considered each collection of best model $F_1$ scores as a sample. Because the Friedman test is nonparametric, we circumvent issues arising from the fact that model performance already causes $F_1$ scores to differ heavily.

With $p < 0.05$, we find the differences between samples to be significant ($p < 0.001$). We then

---

[9]We opted for MBAWO instead of EPM because we were concerned that the stricter EPM metric would overly penalize incremental progress in capturing correct tokens.

[10]For an error analysis and a full catalogue of our results and best model hyperparameters, see Appendices G to I.

| Dataset | Model Components | | Tagging Scheme | | Results | | |
|---|---|---|---|---|---|---|---|
| | Embedding | Encoder | Tagset | Link Type | Precision | Recall | $F_1$ |
| ASP | Learned (Word) | BiLSTM | BIO | Token | 0.16 | 0.13 | 0.14 |
| | word2vec | BiLSTM | BIOMJE | Branch | 0.24 | 0.12 | 0.16 |
| | Learned (Subword) | Transformer | BIOMJ | Token | 0.06 | 0.07 | 0.07 |
| | Latin BERT | – | BIOMJ | Token | **0.51** | 0.33 | **0.40** |
| | Latin BERT | BiLSTM | BIOMJ | Token | 0.47 | **0.34** | 0.39 |
| | Latin BERT | Transformer | BIOME | Branch | 0.44 | 0.31 | 0.37 |
| PSE-I | Chinese BERT | – | BIOM | Branch | 0.30 | 0.42 | 0.35 |
| | Chinese BERT | BiLSTM | BIOMJ | Token | 0.38 | **0.51** | **0.43** |
| | Chinese BERT | Transformer | BIOME | Token | **0.40** | 0.36 | 0.38 |

Table 3: Best test set results, rounded to the second decimal place, on our datasets according to the EPM metric for each Encoder-CRF configuration. The best result in each column is written in boldface.

| Tagset | Link | Avg. Rank ($\downarrow$) |
|---|---|---|
| BIO | Token | 9.17 |
| | Branch | 9.92 |
| BIOE | Token | 9.83 |
| | Branch | 12.75 |
| BIOJ | Token | 11.83 |
| | Branch | 11.08 |
| BIOM | Token | 6.83 |
| | Branch | 6.50 |
| BIOJE | Token | 11.00 |
| | Branch | 13.25 |
| BIOME | Token | 9.17 |
| | Branch | 4.67 |
| BIOMJ | Token | **3.33** |
| | Branch | 4.33 |
| BIOMJE | Token | 5.50 |
| | Branch | 6.83 |

Table 4: Average rank of each tagging scheme's best model for all architectures across the ASP test set. The lowest (*i.e.*, best) result is bolded.

| Embedding + Encoder | $F_1$ (ASP) | $F_1$ (PSE-I) |
|---|---|---|
| Learned (W) + BiLSTM | 0.08 ± 0.03 | – |
| word2vec + BiLSTM | 0.08 ± 0.03 | – |
| Learned (SW) + Transformer | 0.02 ± 0.02 | – |
| BERT | 0.26 ± 0.07 | 0.24 ± 0.07 |
| BERT + BiLSTM | 0.32 ± 0.04 | 0.36 ± 0.04 |
| BERT + Transformer | 0.28 ± 0.05 | 0.30 ± 0.05 |

Table 5: Best test set result averages over each Encoder-CRF configuration for EPM. "W" and "SW" stand for "word" and "subword", respectively. The BERT version used corresponds to each dataset's language.

used a post-hoc Nemenyi test (Nemenyi, 1963) via the `scikit-posthocs` library (Terpilowski, 2019) to determine which tagging scheme pairs achieve significantly different results. With $p < 0.05$, only one pair of tagging schemes significantly differs: BIOMJ-TD and BIOJE-BD—the best and worst schemes, according to the average ranks of each scheme's $F_1$ scores (as presented in Table 4). Given our low sample count, we suspect that further samples might show the superiority of certain schemes. With this in mind, we tentatively recommend any M-based tagging schemes, especially in combination with either of the E or J tags, for use.

In terms of *encoders*, BiLSTMs generally outperformed Transformers. Although non-BiLSTM models achieved peak performance in Table 3, the average performance by BiLSTMs was consistently higher. Table 5 depicts this regardless of the type of embedding used: each BiLSTM model performs

on average better than its Transformer (or encoderless) variant. One possible reason for BiLSTMs' superiority may be that the subtask of predicting distance-based links benefits from LSTMs' ability to count (Weiss et al., 2018; Suzgun et al., 2019b,a).

# 8 Conclusions and Future Work

In this paper, we introduced the task of rhetorical parallelism detection. We described two datasets, an evaluation framework, and several baseline models for the task. Our baselines achieved fairly good performance, and we found that BERT embeddings, a BiLSTM-based encoder, and an M-inclusive tagging scheme were valuable modeling components.

We identify a few directions for future work. The models described here have a closed tagset, so they cannot predict links for distances not seen in the training data; modeling links in other ways might be more effective. Our models only predict flat parallelisms; approaches for nested NER (Finkel and Manning, 2009; Wang et al., 2020) may be a viable direction for extending the Encoder-CRF paradigm toward this end. Finally, applying this work's methods to detect grammatical parallelism might enhance tasks like syntactic parsing, automated essay evaluation, or disfluency detection.

## 9 Limitations

This work introduces a novel task in the form of rhetorical parallelism detection (RPD). Because it is novel, it is innately exploratory. Although it establishes approaches for building and organizing datasets, for evaluating performance at various granularities, and for constructing models to capture parallel structure, it does not do all these perfectly. Thus, it should not be the last word on the topic. In the following paragraphs, we highlight elements of this work which could be improved.

First, this work puts forth two datasets for RPD: the Augustinian Sermon Parallelism (ASP) and the Paibi Student Essay (PSE) (Song et al., 2016) datasets. We annotated both datasets—the former from scratch and the latter on the basis of prior annotations. As is noted in Appendix C, our annotation scheme was not perfect.

- For the ASP dataset, we achieved a 0.4124 EPM $F_1$ score (although higher branch-based and word-based scores) between our two annotators on bootstrapping experiments. Our scores indicate moderate agreement, but the meaningful disagreement implies that our guidelines could be sharpened.

- For the PSE-I dataset—the version of the PSE dataset including our annotations—we did not perform an inter-annotator agreement study. We did not consider it necessary because we were already overlapping with prior annotators' conclusions, but one could argue that the sentence- and span-level annotation tasks differ enough to warrant separate studies.

Second, regarding the ASP dataset, we acknowledge that the use of Latin as a foundation for this task limits its immediate applicability to modern NLP contexts. We believe that the ASP dataset is readily applicable to tasks such as authorship attribution and authorship authentication, which both have precedent in Latin (Forstall and Scheirer, 2010; Kestemont et al., 2016; Kabala, 2020; Corbara et al., 2023) and frequently employ stylistic features in their approaches. Moreover, we believe that ASP can aid distant reading approaches (Moretti, 2000; Jockers, 2013) in permitting the annotation of large corpora with stylistic annotations.

On the other hand, the inclusion of the PSE dataset for RPD provides a modern language for which this task is available. Moreover, late into this work, we became aware of a vein of literature

that could either add to the available Chinese data based on an interest in parallelism detection as a sentence-level or clause-level subtask (Xu et al., 2021; Zhu et al., 2022) as well as work performing narrower, more restricted versions of word-level parallelism detection (Gawryjolek, 2009) or chiasmus detection (Dubremetz and Nivre, 2015, 2018), in English. Future work can potentially expand RPD to the datasets provided by these works.

Third, some modeling decisions were made to suit the needs of our problem and provide baselines without exploring the larger space of possibilities. Namely, as described in Section 6.2 and Appendix D.3, we use a blending function to combine subwords into words so that the appropriate number of tags would be produced. However, we could have also eschewed the CRF altogether and attempted to follow BERT in selecting every word's first tag as its representative tag (Devlin et al., 2019). Future work could further and more deeply investigate the search space of model architectures to improve performance on this task.

## Acknowledgements

We thank Wei Song for working with us to not only provide data used in previous work but also to prepare it for public release. We thank Meng Jiang and Juliana, Joshua, and Joseph Chiang for refining the annotations on the PSE dataset. We thank John Lalor for his suggestions for our inter-annotator agreement study. Finally, we also thank our anonymous reviewers and Brian DuSell, Darcey Riley, Ken Sible, Aarohi Srivastava, and Chihiro Taguchi for their comments and suggestions.

This research was supported in part by an FRSP grant from the University of Notre Dame.

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

## A  Annotation Procedures

In Section 2, we described two major guidelines which directed our study of parallel structures. Here, we list the specific criteria which were used by our annotators to locate parallelisms. Moreover, we describe our annotation scheme in more detail. We begin by enumerating general criteria in Appendix A.1 before discussing specific applications of these criteria in Appendix A.2.

### A.1  General Guidelines

In general, branches of parallelisms were tagged from the first identical (or similar) word to the last identical (or similar) word; as such, each branch was maximally represented. Branches were detected, paired, and combined based upon whether they exhibited at least *two* of these criteria:

- They contained identical number of words (or were within approximately two words of one another).
- They had identical syntactic structure.
- They had two or more pairs of words of an identical grammatical form in identical order.
- They had two or more pairs of words that are lexically identical, are synonyms, are cognates, or are antonyms.
- They had at least two words that have a phonetically similar ending (*e.g.*, rhyme).
- They were a short distance from one another (*i.e.*, they had few intervening words). The acceptable margin was again around two words.

### A.2  Specific Applications

As mentioned earlier, we used the BRAT annotation tool (Stenetorp et al., 2012) to form our datasets.

For the ASP dataset, we developed an annotation scheme consisting of BRAT's entities and relationships. An entity could be a ParallelArm or one of ChiasmA and ChiasmB. A relationship could be Parallel, linking two ParallelArm entities. This relationship indicates a synchystic parallel structure. A relationship could also be Chiasm, linking a ChiasmA and a ChiasmB and labeling a chiastic structure. Our two annotators, both for the initial data collection and the inter-annotator agreement study (see Appendix C), could nest annotations; however, they could not overlap them.

Meanwhile, for the PSE-I dataset, we made a few changes. Because we were mostly interested in annotating sentences that were already marked as parallel in the dataset, we did away with the distinction between synchystic and chiastic parallelisms. Instead, all branches of a parallelism were tagged as Branch entities and connected with Parallel relationships. Annotators were also allowed to use a dummy entity type to tag sentences if they were deemed not to be parallel, but this was never done. The five annotators were told to focus on marked sentences in BRAT, but they were allowed to connect these sentences to unmarked sentences in the same paragraph if they noticed a parallelism.

## B  Additional Metrics

In Section 5, we constructed a framework for defining parallelism-related metrics. We defined one such metric for exact parallelism matches between parallelisms. However, the score and size functions allow for finer granularity. Below, we define three additional metrics which examine parallelisms in terms of their branches, their words-per-branch, and their words altogether.

First, we describe the *maximum parallel branch match* (MPBM) metric. MPBM focuses on branches. It retains a sense of parallel structure by requiring that at least two branches are shared between the hypothesis and reference parallelisms to produce any score. The functions of MPBM are:

$$\mathrm{size}(p) = |p|$$
$$\mathrm{score}(p_1, p_2) = \mathrm{hasParBranch}(p_1, p_2) \cdot |p_1 \cap p_2|$$

where

$$\mathrm{hasParBranch}(q, r) = \mathbb{I}[|q \cap r| \geq 2]$$

Next, we proceed to metrics which examine parallelisms in terms of their words (which we

use interchangeably with tokens below). Let $p$ be a parallelism. We define words$(p) = \bigcup_{(i,j) \in p} \{i, \ldots, j\}$ as the set of all token positions in those branches; this formulation dissolves distinctions between branches for a parallelism. We also define branchedWords$(p) = \{\{i, \ldots, j\} : (i, j) \in p\}$ as the set of all token positions in branches divided up into a set per branch. This formulation preserves branch-level distinctions.

The first word-level metric is the *maximum branch-aware word overlap* (MBAWO) metric. The objective of this metric is similar to MPBM: if at least two distinct branches are matched in a parallelism, then such a matching should obtain some score. However, this metric is less strict than MPBM in that only *words* from two distinct branches need to match. In short, if at least two branches in a hypothesis parallelism have words that match with at least two branches in a reference parallelism, then the match merits a nonzero score based on the number of tokens matched.

To compute this metric, we define $\mathcal{W}(p_1, p_2) =$ branchedWords$(p_1) \times$ branchedWords$(p_2)$; that is, $\mathcal{W}$ pairs all branches of one parallelism with all branches of another parallelism in terms of their word indices. Then we define $\texttt{score}$ and $\texttt{size}$ as:

$$\texttt{size}(p) = |\text{words}(p)|$$
$$\texttt{score}(p_1, p_2) = \max_{W \in \mathcal{W}(p_1, p_2)} \text{hasParWord}(W) \cdot \sum_{(p_1, p_2) \in W} |p_1 \cap p_2|$$

where

$$\text{hasParWord}(W) = \mathbb{I}[\text{wordMatches}(W) \geq 2]$$
$$\text{wordMatches}(W) = \sum_{(p_1, p_2) \in W} \mathbb{I}[|p_1 \cap p_2| > 0]$$

MBAWO calculates an internal maximally-weighted bipartite matching for token overlap; however, it only values those matches which correspond to parallel structure in the reference data. It enforces that through the function wordMatches.

Finally, we define the *maximum word overlap* (MWO) metric. MWO is at the opposite extreme of EPM; it does not necessarily control for the presence of parallel structure. Even so, its ability to measure a model's capacity to find *any* parallel words may be useful. Its functions are as follows:

$$\texttt{size}(p) = |\text{words}(p)|$$
$$\texttt{score}(p_1, p_2) = |\text{words}(p_1) \cap \text{words}(p_2)|$$

## C   Inter-Annotator Agreement Study

To provide some measure of the quality of our data, we performed an inter-annotator agreement study on the ASP dataset. Due to constraints on time and expertise, we recruited our third author, an experienced classicist and Latin teacher, to revisit eight sermons using our annotation scheme. To address the limited annotated data, we computed a bootstrapped estimate for agreement across the whole dataset. After obtaining an initial computation of inter-annotator agreement, we ran 1,000 trials to obtain a 95% confidence interval for our agreement scores. The number of items sampled was equivalent to the total number of matchings made in the initial computation.

Because the space of possible spans and links between spans is prohibitively large, we could not use more traditional inter-annotator agreement metrics such as Cohen's $\kappa$ (Cohen, 1960) or Krippendorff's $\alpha$ (Krippendorff, 2019). Instead, we used our own metrics as defined in Section 5 and Appendix B. However, following the conclusions of Hripcsak and Rothschild that the $F_1$ measure converges to $\kappa$ as the number of negative pairs grows increasingly large, we can treat our $F_1$ scores in a similar manner to chance-corrected agreement statistics.

As a final preprocessing step before matching annotations, we corrected them with two strategies to reduce noise. We felt that these strategies were warranted after discussion unearthed two vague points in our guidelines. We first describe our issues with the guidelines below, and then we clarify the annotation cleaning done afterward.

1. **Conjunctions**: It was unclear whether conjunctions meaningfully contributed to a parallelism. For some parallelisms, they may have been syntactically necessary—but did that mean that they should be part of the parallelism? We decided that conjunctions should be included if *every* branch of the parallelism has one; otherwise, they should be omitted. This allows polysyndeton to be recognized while avoiding the requisite but incidental connections created by some syntactic structure.

2. **Interlocking Parallelisms**: Second, it was difficult to determine the best way to partition parallel structure into branches. This was especially the case with longer parallelisms that involved multiple clauses. Take Fig. 7

| Scoring Object | Actual $F_1$ | Sample Mean and Deviation | Confidence Interval |
|---|---|---|---|
| Parallelism | 0.4615 | $0.4124 \pm 0.0303$ | $[0.4088, 0.4160]$ |
| Branch | 0.5084 | $0.5080 \pm 0.0310$ | $[0.5043, 0.5117]$ |
| Branched Words | 0.5906 | $0.5973 \pm 0.0322$ | $[0.5935, 0.6012]$ |
| Words | 0.6136 | $0.6130 \pm 0.0312$ | $[0.6092, 0.6167]$ |

Table 6: Aggregated actual and sample inter-annotator agreement scores. All values are rounded to the fourth decimal place and range between 0 and 1. Higher is better for the actual $F_1$ and the sample mean; lower is better for the standard deviation. Smaller confidence intervals indicate greater stability of our estimate.

| Latin | *inanis auro,* | | *plenus deo;* | *inanis omni transitoria facultate,* | | *plenus sui domini uoluntate.* |
|---|---|---|---|---|---|---|
| English | [he who] lacks gold, | | [is] full of God; | [he who] lacks every transient supply, | | [is] full of the Lord's own favor. |

| | | | | **Interpretation 1** | | |
|---|---|---|---|---|---|---|
| Stratum 1 | $P_{1,1}$ | | $P_{1,2}$ | $P_{1,3}$ | | $P_{1,4}$ |
| Stratum 2 | – | | – | – | | – |

| | | | | **Interpretation 2** | | |
|---|---|---|---|---|---|---|
| Stratum 1 | | $P_{1,1}$ | | | $P_{1,2}$ | |
| Stratum 2 | $P_{2,1}$ | | $P_{2,2}$ | $P_{3,1}$ | | $P_{3,2}$ |

| | | | | **Interpretation 3** | | |
|---|---|---|---|---|---|---|
| Stratum 1 | | $P_{1,1}$ | | | $P_{1,2}$ | |
| Stratum 2 | – | | – | – | | – |

Figure 7: Selection of possible interpretations of parallel structure of a passage from Augustine's *Sermones* (Sermon 177, Section 4). Clauses are aligned for the Latin and English at the top of the table. Each subsequent section of the table relays an interpretation of parallel structure; each $P_{i,j}$ indicates branch $j$ of parallelism $i$.

as an example. Without clear rules, there are multiple ways to produce parallel structure:

- Interpretation 1 considers each clause its own branch.
- Interpretation 2 considers each pair of clauses, which juxtapose *inanis* ("without, lacking, empty") and *plenus* ("filled, full"), as a single parallelism; moreover, it also recognizes the similar syntactic relationships in the individual clauses.
- Interpretation 3 only annotates the first stratum of the prior interpretation.

Ultimately, we ruled that the third interpretation was best. It *maximally* captured the parallel structure in these clauses. Although it is true that each pair of clauses has its own relationships (*e.g.*, same number of words, same syntactic structure), these relationships also contribute to the larger structure they are contained within, as each *inanis* and *plenus* takes a noun of the same case (ablative) as its object. Thus, these connections are still captured in the third interpretation. We only advise differ-

ing from a maximal interpretation if the parallelisms intentionally use different features.

To address these issues in our original annotations, we automatically altered the annotations on two bases. First, if *all* branches of a parallelism were preceded by a conjunction, then conjunctions were included in all branches; otherwise, no conjunctions were permitted, and any present were removed from annotations. Second, we collapsed interlocking parallelisms into larger branches if they were adjacent. If these larger branches were part of a nested parallelism, and such branches produced a parallelism on an earlier stratum, we removed these larger branches entirely. Although these alterations did cause gains in annotation agreement, they were all between roughly 0.02 to 0.03 $F_1$.

We present our inter-annotator agreement results in Table 6. Across the board, these scores fall below traditional standards (*i.e.*, above 0.67 at the very least) for good chance-corrected reliability in content analysis (Carletta, 1996). However, as noted by Artstein and Poesio, acceptable inter-annotator agreement values remain disputed; the earlier work by Landis and Koch denotes the range between 0.4 and 0.6 as *moderate* agreement and the range

between 0.6 and 0.8 as *substantial* agreement.

Regardless of the exact standard that we follow, we do believe that the disagreement displayed by this inter-annotator agreement study is meaningful and in part due to some vagueness in our original guidelines. We have already updated our guidelines to provide greater specifications for future annotators, and we are currently working on improving our data with these guidelines. We hope to release an enhanced version of our dataset in future work.

## D    Model Idiosyncrasies

In Section 6.2, we go over our general Encoder-CRF model architecture, and we describe some of the major components of our models. Due to space considerations, some details were omitted from that discussion. We discuss such details here.

### D.1    Training <UNK> in BiLSTM-CRFs

In Encoder-CRF models which apply a word-level vocabulary, one necessary issue concerns the handling of unknown words. Usually, such models maintain an <UNK> token for such cases. However, if the <UNK> token is not seen during training, the model will not know how to use it and may confuse it with other tokens seen during training.

Various strategies have been employed to address this problem. Some decide that all words seen less than $k$ times, where $k$ is some preset value, become <UNK> tokens (Reimers and Gurevych, 2017). Others select singletons—words which only appear *once* in the training data—and alter them to become <UNK> during training with a predetermined fixed probability (Lample et al., 2016). We follow the latter strategy, but we augment it so that we can adapt the probability to the dataset at hand and can avoid choosing an arbitrary value.

We propose a singleton replacement probability analogous to Kneser-Ney smoothing (Ney et al., 1994). We compute the frequency of each type in the dataset. Then, we tally the number of types which occur exactly once, $n_1$, and the number of types which occur exactly twice, $n_2$. Here, we use notation from previous work where $n$ represents the number of word types and the subscripts represent a conditional frequency on that count (Chen and Goodman, 1998). Finally, for the singleton replacement probability $p_r$, we use the equation:

$$p_r = \frac{n_1}{n_1 + 2n_2} \tag{1}$$

The Kneser-Ney replacement probability for the ASP dataset, given the current data splits, is approximately 0.6529.

### D.2    Transformer Modifications

We apply the Transformer encoder in some of our models with a slight change from the usual implementation. In line with previous work which sees minor improvements in model performance (Chen et al., 2018; Wang et al., 2019; Nguyen and Salazar, 2019), we swap the traditional order of layer normalization and skip connections. Applying the notation of Nguyen and Salazar, we use PreNorm in the incorporation of the residual connection:

$$\mathbf{x}_{l+1} = \texttt{LayerNorm}(\mathbf{x}_l + F_l(\mathbf{x}_l))$$

where $l$ is the layer's index, $F$ represents the layer itself, and LayerNorm represents the layer normalization operation (Ba et al., 2016).

### D.3    Subword Blending

As previously mentioned, we use WordPiece tokenization with our models which either have a Transformer encoder or Latin BERT embeddings. Using subword-level tokens presents an issue for a word-level tagging problem: in what way should the model process subwords to generate word-level tags? The original BERT paper, which tackles NER, does not use a CRF and instead lets the tag classification of each word's first subword represent the whole word (Devlin et al., 2019).

We take inspiration from BERT's methods and others' approaches to handling subword-to-word relationships (Souza et al., 2019; Casas et al., 2020) by putting a *blending* layer into the architecture. As mentioned in Section 6.2, the blending layer's objective is to combine subword-level encodings into word-level encodings. We define three variations:

- **Take-First**: In this approach, we mimic BERT and select the first subword of each word as its representative. (Note that, when convenient, we abbreviate this variant as "tf".)

- **Mean**: In this approach, we take the mean of all subword encodings per word. This is akin to a subword-aware mean pooling layer.

- **Sum**: In this approach, we sum all subword encodings for a word to represent it. This representation may allow for a model to recognize the accumulation of multiple subwords.

| Dataset | Split | Optimal Ratio | Inner Tag Ratio | Outer Tag Ratio |
|---------|-------|---------------|-----------------|-----------------|
| ASP | Training | 0.7 | 0.7021 | 0.7020 |
| | Validation | 0.1 | 0.0982 | 0.1008 |
| | Optimization | 0.1 | 0.0946 | 0.0980 |
| | Test | 0.1 | 0.1051 | 0.0992 |
| PSE-I | Training | 0.7 | 0.6947 | 0.7007 |
| | Validation | 0.1 | 0.1026 | 0.0998 |
| | Optimization | 0.1 | 0.1005 | 0.0997 |
| | Test | 0.1 | 0.1022 | 0.0998 |

Table 7: Data split ratio results. All ratios are rounded to the fourth decimal place.

| Dataset | Compared Splits | | Inner $p$-Value | Inner Statistic | Outer $p$-Value | Outer Statistic |
|---------|------|------|-----------------|-----------------|-----------------|-----------------|
| ASP | Training | Validation | 0.5797 | 0.5722 | 0.6284 | 0.4990 |
| | Training | Optimization | 0.8347 | −0.2162 | 0.7557 | −0.3236 |
| | Training | Test | 0.7210 | 0.3668 | 0.4727 | 0.7405 |
| | Validation | Optimization | 0.5705 | −0.5816 | 0.5631 | −0.5942 |
| | Validation | Test | 0.8557 | −0.1848 | 0.9601 | 0.0509 |
| | Optimization | Test | 0.6751 | 0.4290 | 0.4980 | 0.7038 |
| PSE-I | Training | Validation | 0.7476 | −0.3236 | 0.6212 | −0.4972 |
| | Training | Optimization | 0.9533 | −0.0588 | 0.8240 | 0.2235 |
| | Training | Test | 0.9882 | −0.0148 | 0.2588 | 1.1381 |
| | Validation | Optimization | 0.8291 | 0.2166 | 0.5779 | 0.5588 |
| | Validation | Test | 0.7987 | 0.2560 | 0.2526 | 1.1540 |
| | Optimization | Test | 0.9720 | 0.0353 | 0.5651 | 0.5780 |

Table 8: Data split Welch's $t$-Test results. All $p$-values and statistics are rounded to the fourth decimal place.

| Dataset | Split | Parallelisms | Branches | Branched Tokens | Tokens |
|---------|-------|--------------|----------|-----------------|--------|
| ASP | Training | 1448 | 3264 | 13833 | 94740 |
| | Validation | 208 | 478 | 1935 | 13580 |
| | Optimization | 191 | 424 | 1863 | 13196 |
| | Test | 215 | 485 | 2070 | 13440 |
| | Total | 2062 | 4651 | 19701 | 134956 |
| PSE-I | Training | 541 | 1478 | 17736 | 168848 |
| | Validation | 87 | 236 | 2618 | 24152 |
| | Optimization | 77 | 219 | 2565 | 24064 |
| | Test | 81 | 220 | 2610 | 24139 |
| | Total | 786 | 2153 | 25529 | 241203 |

Table 9: Data split results per relevant object count. Note that "branched tokens" refers to all tokens in branches—not the number of tokens in general.

# E  Data Splitting Approach

As noted in Section 4, parallelisms are not evenly distributed over sections. Sections are also not evenly distributed over documents. Because we performed evaluation in terms of documents, we wanted to apply a simple heuristic to guarantee that the data splits would fairly distribute tags.

To do this, we used a straightforward approach to optimize split creation. To apply the approach, we supplied it with a set of ratios which govern the quantity and relative size of the splits. We began by taking each file in the dataset and computing counts of its *outer* and *inner* tags.[11] A tag is an outer tag if it is O or M; otherwise, it is an inner tag. Then, we attempted to place a file in each split. If placing that file in a split minimized the mean-squared error (MSE) among the splits, then the file was placed in that split. To be specific, we averaged the MSE for inner and outer tags to compute the final MSE. We repeated this process for every file.

As a further heuristic, we sorted the files after their inner and outer tag counts were computed. We did so by the number of inner tags from maximum

---

[11]As in the main paper, we created *strata* of tags for each dataset equal to the maximum nesting depth of a branch.

to minimum, breaking ties by the number of outer tags. This was in line with our objective to balance out the parallelisms and branches seen across splits: the ordering allowed for files which have a greater effect on MSE to be placed first, thereby letting files with less tags smooth out error gradually.

To verify our procedure, we provide the resulting mean-squared error for our splits. We also used Welch's $t$-test for this purpose. For each pair of splits created, we ran this test twice with per-file counts of the inner and outer tags. In this case, we do *not* want the test to report significance; rather, in showing that the difference between the distributions is insignificant (*e.g.*, $p \geq 0.05$), we are showing that the splits are divided up such that they could have been drawn from the same distribution. We use the SciPy implementation of Welch's $t$-test for this purpose (Virtanen et al., 2020).

# F  Hyperparameter Search Spaces

For our hyperparameter search experiments in Section 7.2, we performed a random hyperparameter search (Bergstra and Bengio, 2012) of eight trials per setting. However, because we are proposing a new task, we did not have prior literature on successful hyperparameters which we could draw upon in order to define hyperparameter spaces. Thus, to perform these experiments, we created our own.

These hyperparameter search experiments were governed by three main elements:

1. The set of hyperparameters which are selected for variation. Although many hyperparameters could be varied, only some were chosen to direct the search toward hyperparameters which were suspected to be meaningful.

2. The space itself from which hyperparameters were drawn, as this determines all possible values which the hyperparameter can take.

3. The set of constraints which are imposed upon the random trials. Naturally, we want to prevent any trial from being duplicated; however, there are other inductive biases which may guide our search.

In the subsections below, we expound upon each of these elements and provide tables which summarize pertinent information regarding them.

## F.1  Selected Hyperparameters

Table 11 provides information about what hyperparameters are varied per model. Due to space considerations, we use a number of abbreviations; these are supplied in Table 10.

## F.2  Designated Hyperparameter Spaces

Table 12 lists the hyperparameter spaces which we used during the random search process. While some of the above hyperparameters are relatively self-explanatory in their functionality, we provide an explanation for those which may be less clear or otherwise warrant further explanation:

- Activation Function: The PyTorch (Paszke et al., 2019) implementation of the Transformer (Vaswani et al., 2017) provides integrations for both ReLU and GeLU (Hendrycks and Gimpel, 2016) as activation functions. We uphold that support in our modifications (as presented in Appendix D.2) to see whether they have any major effects on our results.

- Dimensionalities: For all embedding and hidden dimensionality spaces, we let the search space be in powers of two and at every value averaged between adjacent powers of two. We used this approach to explore a decent portion of the search space without leaving big gaps between larger powers. Furthermore, we differed the spaces for BiLSTMs and Transformers to adapt more to precedent while still allowing some exploration. In particular, we took inspiration from some hyperparameter LSTM searches in NER (Reimers and Gurevych, 2017) and the general application of the Transformer (Vaswani et al., 2017).

- Depth: In initial trials, we found that high depth values prevented the models from learning meaningful information. Thus, instead of setting a maximum of 6 encoder layers as per the initial Transformer implementation (Vaswani et al., 2017), we lowered it to 4.

- Learning Rates: In the NER literature, a variety of learning rates have been used (alongside a variety of optimizers). Values from .1 (Huang et al., 2015) to .015 (Ma and Hovy, 2016) to .01 (Lample et al., 2016) to a variety of manually-tuned others (Reimers and Gurevych, 2017) have been proposed. Due to the differences in our task, however, we examine a wider range on the basis of those

| Abbreviation | Full Hyperparameter Name |
| --- | --- |
| AF | Activation Function |
| BF | Blending Function |
| HS | Hidden Size |
| IS | Input Size |
| LR | Learning Rate |
| Lemma | Lemmatization |

Table 10: List of all hyperparameter abbreviations.

| Embedding | Encoder | Varied Hyperparameters |
| --- | --- | --- |
| Learned (Word) | BiLSTM | Depth, HS, IS, LR, Lemma |
| word2vec | BiLSTM | Depth, HS, LR |
| Learned (Subword) | Transformer | AF, BF, Depth, Heads, HS, IS, LR |
| (Chinese/Latin) BERT | – | BF, LR |
| (Chinese/Latin) BERT | BiLSTM | BF, Depth, HS, LR |
| (Chinese/Latin) BERT | Transformer | AF, BF, Depth, Heads, HS, LR |

Table 11: Listing of all hyperparameters varied per model.

| Hyperparameter | Space |
| --- | --- |
| Activation Function | {ReLU, GeLU} |
| Blender | {mean, sum, take-first} |
| Depth | {1, 2, 3, 4} |
| Heads | {1, 2, 4, 8} |
| Hidden Size (BiLSTM) | {32, 48, 64, 96, 128, 192, 256, 384, 512} |
| Hidden Size (Transformer) | {256, 384, 512, 768, 1024, 1280, 1536, 1792, 2048} |
| Input Size (BiLSTM) | {64, 96, 128, 192, 256, 384, 512, 768} |
| Input Size (Transformer) | {128, 192, 256, 384, 512, 768, 1024} |
| Learning Rate | {0.0001, 0.0002, . . . , 0.01} |
| Lemmatization | {True, False} |

Table 12: Collection of all hyperparameter search spaces.

values. Previous experience in using Adam (Kingma and Ba, 2015) has seen models struggle with high learning rates, hence the choice of varying between lower ones.

For all such hyperparameter search spaces, we sampled from each uniformly.

### F.3 Hyperparameter Trial Constraints

Although we could theoretically generate any combination of hyperparameters in our randomly-sought trials, it is not necessarily the case that any combination will be fruitful. With a wide enough search space, it is possible that random search will avoid discovering regions of the search space where performance is approximately maximal. As a re-

sult, we direct the random search process in some cases by forcing trials to meet a set of constraints.

The main constraint we supplied was for the BiLSTM-based models. This constraint was the *word-level dimensionality compression* constraint, which assured that HS ≤ IS. The hidden state is intended here to compress and learn how to use word embedding information. This constraint is applied implicitly for the model with BERT embeddings and the BiLSTM encoder, as every dimensionality in the search space was at or below 768.

### G Error Analysis

The main body of this paper, and especially Section 7.2, highlights major hyperparameters for our

|  | Parallelism | | Branch | | | Word | | | Total |
|  | FP | FN | FP | FN | FM | FP | FN | FM | (by Sermon) |
|---|---|---|---|---|---|---|---|---|---|
| Sermon 24 | 8 | 22 | 3 | 3 | 2 | 0 | 2 | 0 | 40 |
| Sermon 175 | 7 | 13 | 0 | 3 | 0 | 1 | 1 | 0 | 25 |
| Sermon 177 | 11 | 39 | 2 | 2 | 1 | 0 | 1 | 0 | 56 |
| Sermon 188 | 7 | 7 | 2 | 0 | 0 | 0 | 4 | 0 | 20 |
| Sermon 207 | 7 | 4 | 2 | 0 | 0 | 1 | 0 | 0 | 14 |
| Sermon 211 | 4 | 19 | 0 | 1 | 0 | 0 | 1 | 0 | 25 |
| Sermon 219 | 0 | 9 | 0 | 0 | 0 | 0 | 0 | 0 | 9 |
| Sermon 222 | 1 | 1 | 0 | 0 | 0 | 0 | 0 | 0 | 2 |
| Sermon 271 | 0 | 5 | 0 | 0 | 0 | 0 | 0 | 0 | 5 |
| Total (by Category) | 45 | 119 | 9 | 9 | 3 | 2 | 9 | 0 | 196 |

Table 13: A presentation of our error analysis categorization results on ASP's validation set with the model with the highest $F_1$ score from Section 7.2.

Encoder-CRF architecture which improve performance. In this section, we attempt to provide some insight about where our Encoder-CRF models could still improve by describing the kinds of errors that these models make. We focus on the ASP dataset in this section, as this dataset was designed for word-level parallelism detection.

To elucidate relevant and constructive model behavior, we defined a set of categories for this error analysis. That set of categories can be divided into a three-by-three grid on the basis of three error *granularities* and three error *types*. Following our metrics, these granularities are at the levels of parallelisms, branches, and words. Meanwhile, two of our error types derive from classification error types: *false positive* (FP) and *false negative* (FN).

Our third error type, *false mixture* (FM), is a combination of the FP and FN which applies to branches and words. Because a model can both predict a branch or word that a parallelism does not contain (false positive) and omit a branch or word that a parallelism contains (false negative), both error types are simultaneously possible for these granularities. Conversely, a parallelism cannot both be entirely missing and entirely distinct from reference data; thus, the granularity-type combination of parallelism and false mixture is not possible. This leaves us with a set of eight categories.

We went over ASP's validation set with this annotation scheme, comparing hypothesized parallelisms from the model with the highest-scoring $F_1$ score from Section 7.2 and reference parallelisms. We added a category for every matched hypothesis parallelism and reference parallelism. In other words, a hypothesis and reference are only categorized once if they have some overlap; otherwise, they are categorized independently. We provide a more detailed, category-by-category overview of this scheme in our main code repository.

We present the results of our error analysis categorization in Table 13. From this table, we see that the predominant cause of error comes from the highest level of granularity: parallelisms. Conversely, we see that other kinds of error are quite infrequent. As a result, we can surmise that these Encoder-CRF models are relatively all-or-nothing in their sequence labeling approach. They are proficient at getting entire parallelisms correct, and largely down to the boundaries, when they detect a parallelism. However, their ability to confidently detect them in full needs more work.

To illustrate some of these error categories, we provide a set of three examples in Fig. 8. Fig. 8(a) presents the most common case: the false negative parallelism. Fig. 8(b), in turn, presents its false positive version. Finally, Fig. 8(c) shows what a false mixture looks like.

Starting with Fig. 8(a), we find a two-branch parallelism from ASP. Both branches contain two clauses spanning four words, are headed by subjects of the same case and number, and apply verbal forms in their first clauses with the same lemma (*promitto*). Augustine plays up contrasts by juxtaposing human beings and God as well as verbs for belief (*crederes*) and doubt (*dubitas*). The moods used to define the conditional structures of each branch, being contrafactual and factual, respectively, further highlight this contrast. In short, these

(a) [ *homo* *si promitteret* , *crederes* ]₁ ; [ *deus*
human:NOM;SG if promise:IPFV;ACT;SBJV;3SG , believe:IPFV;ACT;SBJV;2SG ; god:NOM;SG
*promittit* , *et* *dubitas* ]₁ ?
promise:PRS;ACT;IND;3SG , and doubt:PRS;ACT;IND;2SG ?
"If a human were promising, you would believe them; God promises, and you doubt Him?"

(b) *... quam illa quae* ⟨ *caeli* *creatorem* *de* *caelo* *deposuit* ⟩₁ , *et* ⟨
... than that who heaven:GEN;SG maker:ACC;SG from sky:ABL;SG put.down:PRF;ACT;IND;3SG , and
*terreno* *corpore* *terrae* *induit* *conditorem* ⟩₁ ; *...*
earthen:ABL;SG body:ABL;SG earth:GEN;SG clothe:PRF;ACT;IND;3SG builder:ACC;SG ; ...
"... than that which put the maker of heaven down from heaven, and that which clothed the builder of the earth with an earthen body; ..."

(c) *fecit enim hoc* [ *per* *fideles* *suos* ]₁ , [ ⟨ *per* *christianos* *suos* ⟩₁]₁ , ⟨ *per*
do for this through faithful:ACC;PL own:ACC;PL , through christian:ACC;PL own:ACC;PL , through
*potestates* ⟩₁ *a se* *ordinatas et christi iugo iam* *subditas* .
authority:ACC;PL by oneself govern and christ yoke already subject .
"For he did this through his own faithful, through his own Christians, through the authorities governed by him and subjected already to the yoke of Christ."

Figure 8: Examples of errors for the ASP dataset's validation set and the model with the highest $F_1$ score from Section 7.2. Blue square brackets demarcate branches in the reference data, red triangular brackets demarcate branches in the hypothesis data, and numbering indicates the parallelism to which a branch belongs relative to its data source. Below each Latin sentence, we provide a word-by-word English translation, gloss some morphological features in branches, and present an idiomatic translation. Example (a) is from Sermon 177, Section 11 and displays a parallelism-level false negative. Example (b) is from Sermon 207, Section 1 and displays a parallelism-level false positive. Example (c) is from Sermon 24, Section 7 and displays a branch-level false mixture.

clauses contain many distinct cues as to their parallel nature. In spite of this, our model did not detect this pair. One possibility is that variations on token order due to function words (*si*, *et*) and commas caused the model to ignore the branch pair.

Next, Fig. 8(b) shows a predicted parallelism that the reference data did not contain. On the one hand, there are aspects of this branch pair that make them seem parallel. The clauses juxtapose heaven (*caeli*, *de caelo*) and earth (*terreno*, *terrae*) in the same grammatical roles. Both also use verbs conjugated in the same manner, and each takes a direct object in the same case (*creatorem*, *conditorem*) which are synonymous. On the other hand, the word order in these clauses calls their parallel nature into question; there is no clear alignment between the word orders in each clause (*e.g.*, the direct objects are the second and fifth words, respectively). As a result, it seems understandable both why the model was fooled by this sentence as well as why it was not annotated in the first place.

Finally, we provide an example of a branch-level false mixture in Fig. 8(c). In this branch-level false mixture, the model agrees with the reference parallelism on its second branch. However, it both ignores the reference's first branch and adds an additional branch after the second. The reference contains a relatively simple parallelism consisting of two three-word branches centering around two prepositional phrases that share the word *suos*. The pair are followed by a third clause that is not part of the parallelism due to its rather elaborate content and lack of *suos* (although, it contains the related form *se*). To speculate about the model's mistake, it may have been drawn by lexical connections between the second branch and the final clause (*i.e.*, *per* with *per*, *suos* with *se*, and *christianos* with *christi*). However, because of the final clause's additional complexity, it then only selected a couple words to pair with the second branch.

## H Exhaustive Experimental Results

In this section, we display box plots which give an overview of all $F_1$ scores attained from each of the best models across all configurations explored during the hyperparameter search and all metrics. All box plots are created from 16 data points, as each model architecture variation (*i.e.*, embedding-encoder combination) was examined across sixteen distinct tagging schemes. The ASP dataset's plots are presented in Figs. 9 to 12. Meanwhile, the PSE-I dataset's plots are presented in Figs. 13 to 16. We also provide tabular data and CSV data for all our results, additionally including precision and recall for all metrics, in our main code repository.

# I Exemplary Model Hyperparameter Tables

In this section, we provide sets of tables which describe the hyperparameters used for each best trial catalogued in Appendix H. For the ASP dataset, we organized the results into three tables—Tables 14 to 16—based upon what hyperparameters were varied. For the PSE-I dataset, we organized them similarly in Tables 17 and 18.

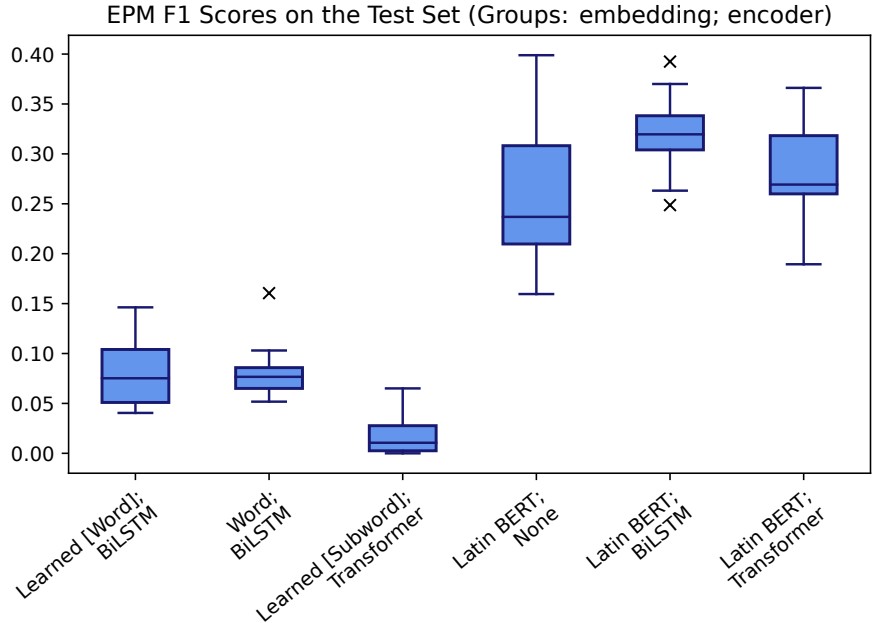

Figure 9: Box plots presenting model-relative results on the ASP dataset's test set using the Exact Parallelism Match (EPM) metric.

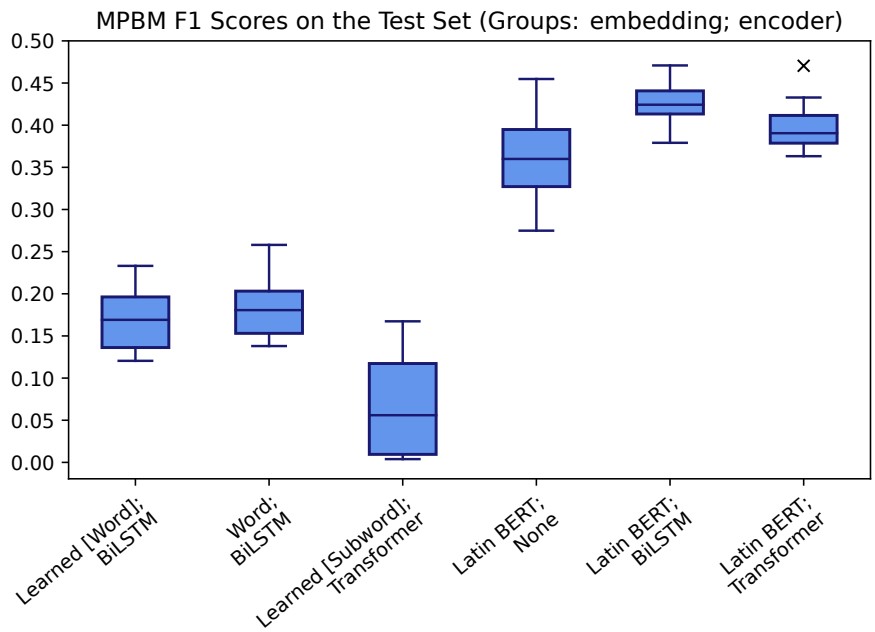

Figure 10: Box plots presenting model-relative results on the ASP dataset's test set using the Maximum Parallel Branch Match (MPBM) metric.

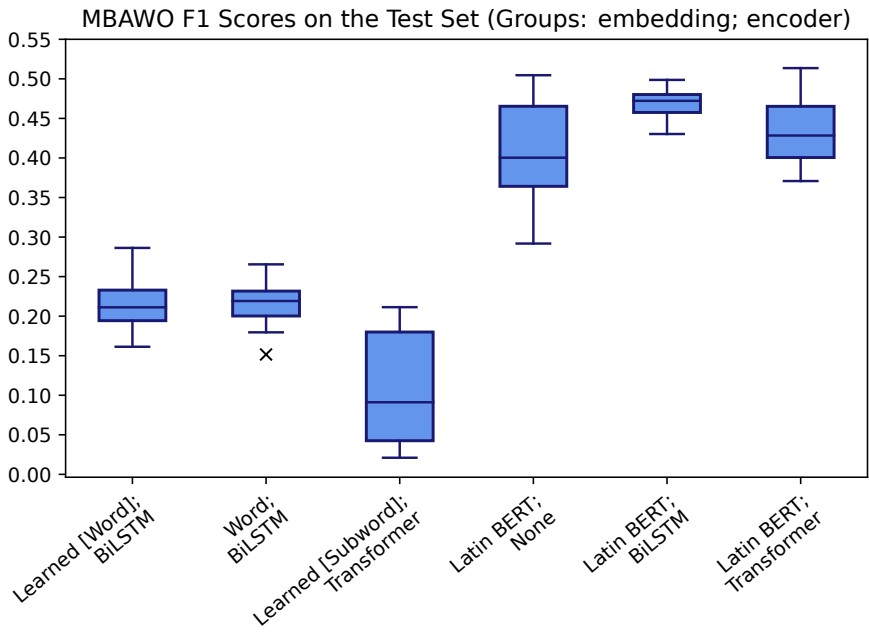

Figure 11: Box plots presenting model-relative results on the ASP dataset's test set using the Maximum Branch-Aware Word Overlap (MBAWO) metric.

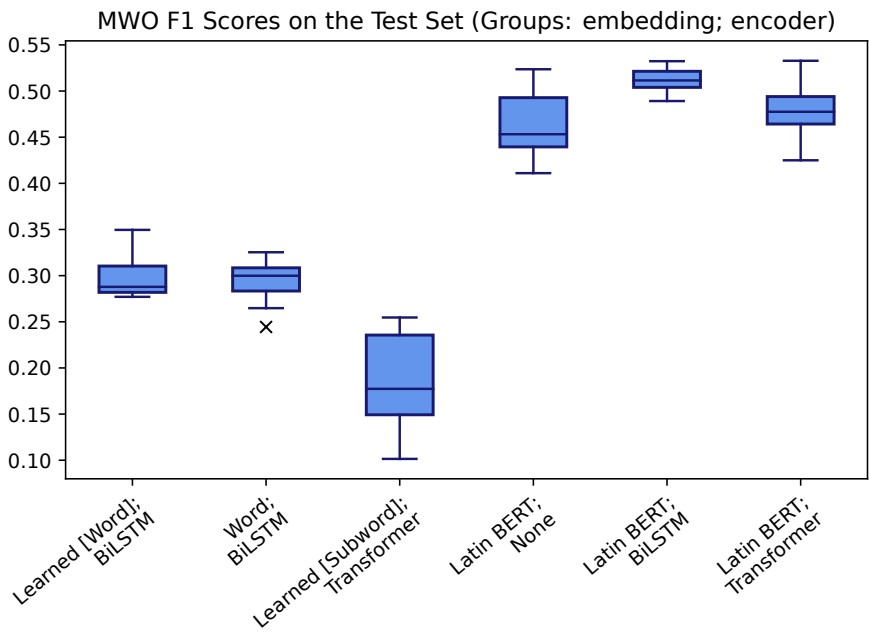

Figure 12: Box plots presenting model-relative results on the ASP dataset's test set using the Maximum Word Overlap (MWO) metric.

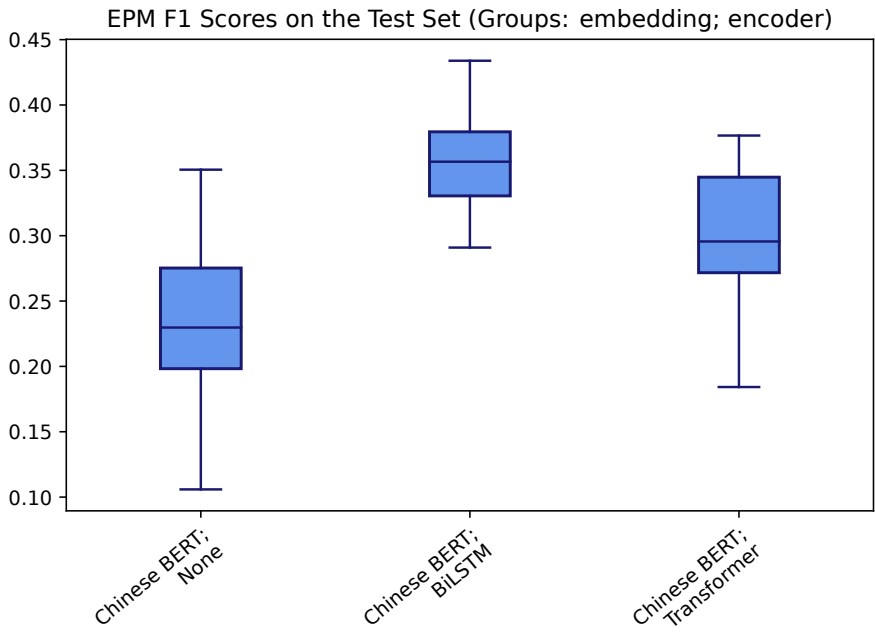

Figure 13: Box plots presenting model-relative results on the PSE-I dataset's test set using the Exact Parallelism Match (EPM) metric.

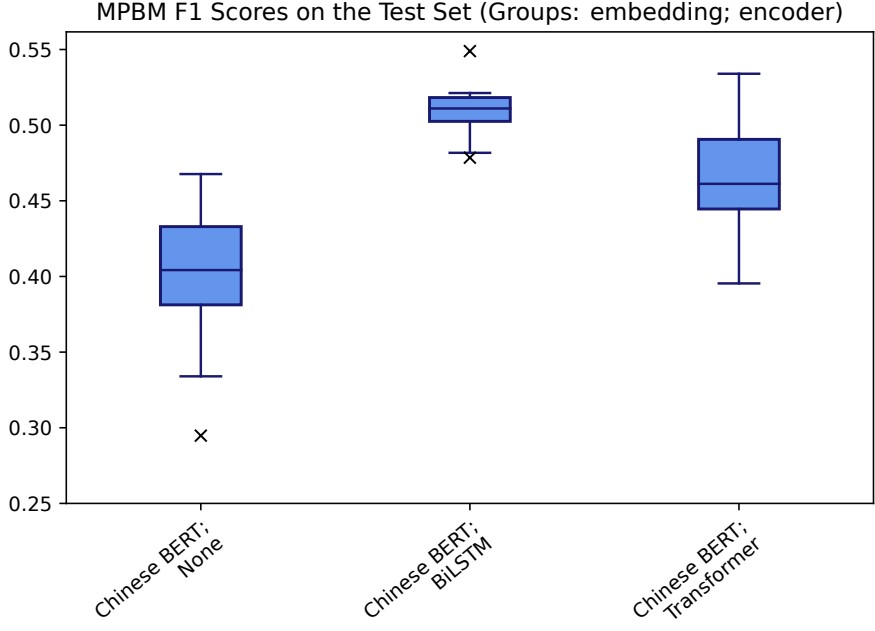

Figure 14: Box plots presenting model-relative results on the PSE-I dataset's test set using the Maximum Parallel Branch Match (MPBM) metric.

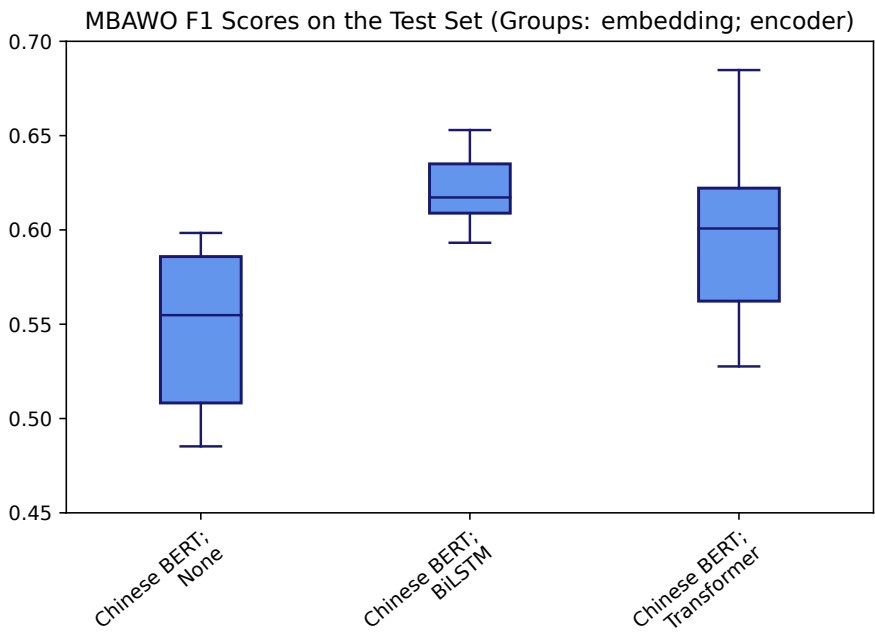

Figure 15: Box plots presenting model-relative results on the PSE-I dataset's test set using the Maximum Branch-Aware Word Overlap (MBAWO) metric.

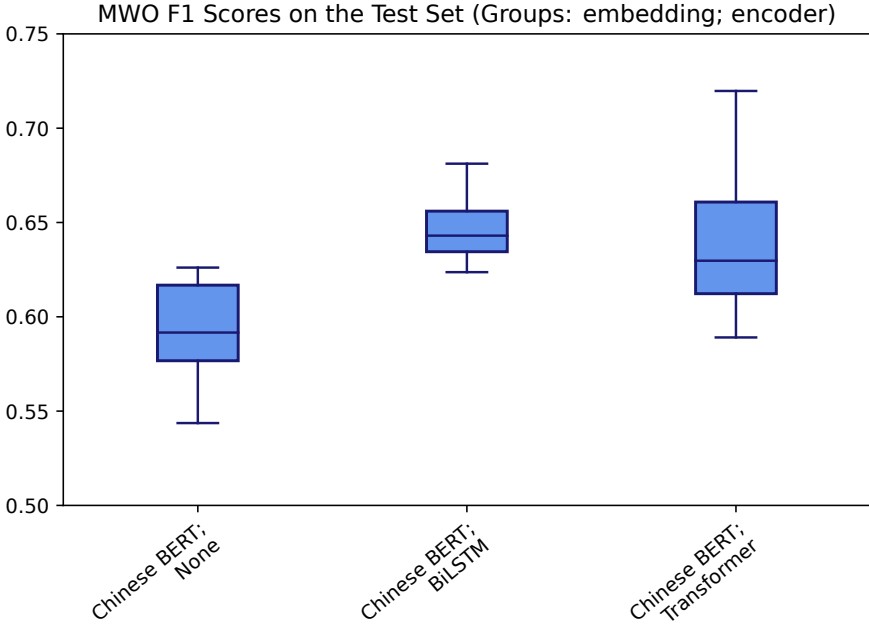

Figure 16: Box plots presenting model-relative results on the PSE-I dataset's test set using the Maximum Word Overlap (MWO) metric.

| Model Components | | Tagging Scheme | | Hyperparameters | | | | |
|---|---|---|---|---|---|---|---|---|
| Embedding | Encoder | Tagset | Link Type | DEPTH | HS | IS | LEMMA | LR |
| Learned (Word) | BiLSTM | BIO | Token | 3 | 384 | 512 | N | 0.0013 |
| | | | Branch | 4 | 384 | 768 | N | 0.0008 |
| | | BIOE | Token | 4 | 64 | 768 | N | 0.0021 |
| | | | Branch | 3 | 48 | 256 | N | 0.0039 |
| | | BIOJ | Token | 1 | 64 | 512 | N | 0.0076 |
| | | | Branch | 3 | 384 | 384 | N | 0.0028 |
| | | BIOM | Token | 4 | 128 | 384 | N | 0.0010 |
| | | | Branch | 1 | 128 | 128 | N | 0.0035 |
| | | BIOJE | Token | 4 | 256 | 768 | N | 0.0063 |
| | | | Branch | 2 | 256 | 256 | Y | 0.0034 |
| | | BIOME | Token | 4 | 32 | 192 | N | 0.0045 |
| | | | Branch | 4 | 128 | 256 | N | 0.0082 |
| | | BIOMJ | Token | 4 | 96 | 512 | N | 0.0052 |
| | | | Branch | 2 | 64 | 768 | N | 0.0047 |
| | | BIOMJE | Token | 3 | 192 | 512 | Y | 0.0018 |
| | | | Branch | 3 | 128 | 192 | Y | 0.0085 |
| word2vec | BiLSTM | BIO | Token | 4 | 96 | 300 | Y | 0.0028 |
| | | | Branch | 3 | 256 | 300 | Y | 0.0040 |
| | | BIOE | Token | 3 | 32 | 300 | Y | 0.0076 |
| | | | Branch | 3 | 128 | 300 | Y | 0.0029 |
| | | BIOJ | Token | 3 | 48 | 300 | Y | 0.0072 |
| | | | Branch | 4 | 64 | 300 | Y | 0.0059 |
| | | BIOM | Token | 3 | 256 | 300 | Y | 0.0028 |
| | | | Branch | 4 | 256 | 300 | Y | 0.0059 |
| | | BIOJE | Token | 3 | 32 | 300 | Y | 0.0035 |
| | | | Branch | 4 | 48 | 300 | Y | 0.0049 |
| | | BIOME | Token | 4 | 192 | 300 | Y | 0.0002 |
| | | | Branch | 3 | 32 | 300 | Y | 0.0032 |
| | | BIOMJ | Token | 1 | 256 | 300 | Y | 0.0035 |
| | | | Branch | 4 | 96 | 300 | Y | 0.0058 |
| | | BIOMJE | Token | 3 | 128 | 300 | Y | 0.0022 |
| | | | Branch | 4 | 192 | 300 | Y | 0.0028 |

Table 14: Best hyperparameters for non-BERT models with a BiLSTM encoder for the ASP dataset.

| Model Components | | Tagging Scheme | | Hyperparameters | | | | | | |
|---|---|---|---|---|---|---|---|---|---|---|
| Embedding | Encoder | Tagset | Link Type | AF | BF | Depth | Heads | HS | IS | LR |
| Learned (Subword) | Transformer | BIO | Token | ReLU | sum | 3 | 4 | 384 | 192 | 0.0081 |
| | | | Branch | ReLU | mean | 1 | 4 | 768 | 128 | 0.0076 |
| | | BIOE | Token | ReLU | tf | 1 | 1 | 1280 | 192 | 0.0071 |
| | | | Branch | GeLU | tf | 2 | 8 | 384 | 768 | 0.0095 |
| | | BIOJ | Token | GeLU | tf | 2 | 4 | 512 | 1024 | 0.0052 |
| | | | Branch | GeLU | mean | 1 | 8 | 2048 | 768 | 0.0043 |
| | | BIOM | Token | ReLU | tf | 2 | 8 | 1792 | 192 | 0.0011 |
| | | | Branch | GeLU | mean | 1 | 4 | 2048 | 256 | 0.0060 |
| | | BIOJE | Token | GeLU | sum | 2 | 2 | 512 | 768 | 0.0082 |
| | | | Branch | ReLU | mean | 2 | 1 | 384 | 128 | 0.0051 |
| | | BIOME | Token | GeLU | sum | 4 | 8 | 1280 | 384 | 0.0071 |
| | | | Branch | GeLU | sum | 2 | 8 | 384 | 384 | 0.0056 |
| | | BIOMJ | Token | ReLU | sum | 3 | 2 | 512 | 384 | 0.0065 |
| | | | Branch | GeLU | tf | 2 | 2 | 384 | 512 | 0.0100 |
| | | BIOMJE | Token | ReLU | mean | 4 | 4 | 256 | 192 | 0.0087 |
| | | | Branch | GeLU | sum | 3 | 8 | 1280 | 512 | 0.0100 |
| Latin BERT | Transformer | BIO | Token | GeLU | sum | 1 | 8 | 256 | 768 | 0.0090 |
| | | | Branch | ReLU | mean | 4 | 1 | 1024 | 768 | 0.0086 |
| | | BIOE | Token | ReLU | sum | 4 | 4 | 1280 | 768 | 0.0028 |
| | | | Branch | ReLU | sum | 3 | 2 | 768 | 768 | 0.0034 |
| | | BIOJ | Token | ReLU | mean | 4 | 1 | 1792 | 768 | 0.0007 |
| | | | Branch | GeLU | mean | 4 | 1 | 1024 | 768 | 0.0021 |
| | | BIOM | Token | GeLU | sum | 3 | 2 | 1024 | 768 | 0.0035 |
| | | | Branch | GeLU | sum | 3 | 2 | 1024 | 768 | 0.0035 |
| | | BIOJE | Token | GeLU | mean | 2 | 1 | 1792 | 768 | 0.0048 |
| | | | Branch | GeLU | mean | 2 | 4 | 1536 | 768 | 0.0037 |
| | | BIOME | Token | GeLU | mean | 4 | 4 | 1024 | 768 | 0.0016 |
| | | | Branch | GeLU | sum | 2 | 2 | 1536 | 768 | 0.0033 |
| | | BIOMJ | Token | ReLU | tf | 3 | 1 | 384 | 768 | 0.0061 |
| | | | Branch | ReLU | mean | 2 | 1 | 1536 | 768 | 0.0014 |
| | | BIOMJE | Token | ReLU | mean | 4 | 1 | 256 | 768 | 0.0086 |
| | | | Branch | GeLU | mean | 3 | 8 | 384 | 768 | 0.0045 |

Table 15: Best hyperparameters for models with a Transformer encoder on the ASP dataset.

| Model Components | | Tagging Scheme | | Hyperparameters | | | |
|---|---|---|---|---|---|---|---|
| Embedding | Encoder | Tagset | Link Type | BF | DEPTH | HS | LR |
| Latin BERT | – | BIO | Token | sum | – | – | 0.0055 |
| | | | Branch | tf | – | – | 0.0010 |
| | | BIOE | Token | sum | – | – | 0.0012 |
| | | | Branch | sum | – | – | 0.0055 |
| | | BIOJ | Token | tf | – | – | 0.0077 |
| | | | Branch | sum | – | – | 0.0070 |
| | | BIOM | Token | tf | – | – | 0.0064 |
| | | | Branch | sum | – | – | 0.0012 |
| | | BIOJE | Token | sum | – | – | 0.0063 |
| | | | Branch | mean | – | – | 0.0005 |
| | | BIOME | Token | mean | – | – | 0.0045 |
| | | | Branch | mean | – | – | 0.0021 |
| | | BIOMJ | Token | tf | – | – | 0.0005 |
| | | | Branch | mean | – | – | 0.0030 |
| | | BIOMJE | Token | sum | – | – | 0.0001 |
| | | | Branch | mean | – | – | 0.0061 |
| Latin BERT | BiLSTM | BIO | Token | mean | 3 | 384 | 0.0007 |
| | | | Branch | sum | 3 | 96 | 0.0022 |
| | | BIOE | Token | tf | 4 | 48 | 0.0024 |
| | | | Branch | mean | 3 | 512 | 0.0012 |
| | | BIOJ | Token | mean | 3 | 256 | 0.0008 |
| | | | Branch | sum | 4 | 128 | 0.0066 |
| | | BIOM | Token | sum | 2 | 192 | 0.0048 |
| | | | Branch | tf | 3 | 192 | 0.0073 |
| | | BIOJE | Token | mean | 4 | 48 | 0.0017 |
| | | | Branch | sum | 3 | 128 | 0.0010 |
| | | BIOME | Token | mean | 2 | 64 | 0.0083 |
| | | | Branch | sum | 4 | 128 | 0.0026 |
| | | BIOMJ | Token | sum | 3 | 96 | 0.0028 |
| | | | Branch | tf | 4 | 192 | 0.0013 |
| | | BIOMJE | Token | sum | 4 | 192 | 0.0055 |
| | | | Branch | mean | 2 | 96 | 0.0086 |

Table 16: Best hyperparameters for models with BERT-based embeddings and a non-Transformer encoder on the ASP dataset.

| Model Components | | Tagging Scheme | | | | | | | | |
|---|---|---|---|---|---|---|---|---|---|---|
| | | | | | | Hyperparameters | | | | |
| Embedding | Encoder | Tagset | Link Type | AF | BF | Depth | Heads | HS | IS | LR |
| Chinese BERT | Transformer | BIO | Token | ReLU | sum | 2 | 8 | 1280 | 768 | 0.0010 |
| | | | Branch | ReLU | sum | 2 | 2 | 1792 | 768 | 0.0040 |
| | | BIOE | Token | ReLU | tf | 1 | 1 | 256 | 768 | 0.0088 |
| | | | Branch | GeLU | tf | 2 | 2 | 1280 | 768 | 0.0010 |
| | | BIOJ | Token | ReLU | tf | 4 | 8 | 1536 | 768 | 0.0063 |
| | | | Branch | ReLU | tf | 1 | 1 | 512 | 768 | 0.0078 |
| | | BIOM | Token | GeLU | tf | 4 | 1 | 256 | 768 | 0.0100 |
| | | | Branch | ReLU | mean | 1 | 8 | 1792 | 768 | 0.0094 |
| | | BIOJE | Token | GeLU | tf | 2 | 4 | 1536 | 768 | 0.0016 |
| | | | Branch | ReLU | sum | 1 | 2 | 1536 | 768 | 0.0061 |
| | | BIOME | Token | GeLU | mean | 2 | 4 | 2048 | 768 | 0.0001 |
| | | | Branch | GeLU | mean | 1 | 8 | 1536 | 768 | 0.0086 |
| | | BIOMJ | Token | GeLU | mean | 2 | 4 | 768 | 768 | 0.0025 |
| | | | Branch | ReLU | mean | 3 | 2 | 1280 | 768 | 0.0076 |
| | | BIOMJE | Token | ReLU | mean | 3 | 4 | 512 | 768 | 0.0073 |
| | | | Branch | GeLU | sum | 3 | 4 | 384 | 768 | 0.0020 |

Table 17: Best hyperparameters for models with a Transformer encoder on the PSE-I dataset.

| Model Components | | Tagging Scheme | | Hyperparameters | | | |
|---|---|---|---|---|---|---|---|
| Embedding | Encoder | Tagset | Link Type | BF | DEPTH | HS | LR |
| Chinese BERT | – | BIO | Token | `tf` | – | – | .0002 |
| | | | Branch | mean | – | – | .0089 |
| | | BIOE | Token | mean | – | – | .0063 |
| | | | Branch | mean | – | – | .0012 |
| | | BIOJ | Token | mean | – | – | .0043 |
| | | | Branch | sum | – | – | .0072 |
| | | BIOM | Token | sum | – | – | .0020 |
| | | | Branch | mean | – | – | .0006 |
| | | BIOJE | Token | mean | – | – | .0050 |
| | | | Branch | `tf` | – | – | .0028 |
| | | BIOME | Token | mean | – | – | .0087 |
| | | | Branch | `tf` | – | – | .0037 |
| | | BIOMJ | Token | mean | – | – | .0092 |
| | | | Branch | `tf` | – | – | .0017 |
| | | BIOMJE | Token | `tf` | – | – | .0052 |
| | | | Branch | mean | – | – | .0066 |
| Chinese BERT | BiLSTM | BIO | Token | mean | 4 | 192 | .0012 |
| | | | Branch | sum | 2 | 96 | .0019 |
| | | BIOE | Token | sum | 3 | 32 | .0025 |
| | | | Branch | sum | 2 | 64 | .0043 |
| | | BIOJ | Token | sum | 2 | 512 | .0002 |
| | | | Branch | mean | 1 | 48 | .0007 |
| | | BIOM | Token | mean | 3 | 128 | .0080 |
| | | | Branch | mean | 2 | 32 | .0056 |
| | | BIOJE | Token | mean | 4 | 64 | .0040 |
| | | | Branch | sum | 4 | 128 | .0003 |
| | | BIOME | Token | `tf` | 4 | 32 | .0016 |
| | | | Branch | `tf` | 4 | 32 | .0010 |
| | | BIOMJ | Token | sum | 3 | 128 | .0004 |
| | | | Branch | sum | 4 | 48 | .0011 |
| | | BIOMJE | Token | `tf` | 1 | 32 | .0058 |
| | | | Branch | mean | 2 | 32 | .0025 |

Table 18: Best hyperparameters for models with BERT-based embeddings and a non-Transformer encoder on the PSE-I dataset.