# OpenReview forum: "Introducing Rhetorical Parallelism Detection: A New Task with Datasets, Metrics, and Baselines"
_EMNLP/2023/Conference — EMNLP 2023 Main_

### Official Review · Reviewer_MynM · 2023-07-27

**Soundness:** 5

**Excitement:**

4: Strong: This paper deepens the understanding of some phenomenon or lowers the barriers to an existing research direction.

**Paper Topic And Main Contributions:**

This paper proposes rhetorical parallelism detection (RPD) as a novel NLP task in which the objective is to identify sets of textual spans in natural language that constitute a parallelism as defined by rhetoreticians. To this end, the authors formalize the task and several plausible evaluation metrics, offer a novel expert-annotated dataset of naturally-occurring parallelisms in Latin and Chinese texts, identify and suggest solutions to important technical challenges (e.g., which tagging scheme to use), and report the performance of a diverse array of baseline models, the results of which show both substantial evidence of learning but also much potential for improvement, suggesting that this is a tractable but challenging task.

I'm impressed by this paper and support publication. I do have some concerns that I elaborate on below.

**Reasons To Accept:**

1. Clear problem statement, motivation, methods, results, and discussion
2. Plausibly broad interest, especially for applications of NLP to writing pedagogy
3. Impressive scope of work, including a novel task and formalization, a large expert-annotated resource, and extensive modeling
4. Methods and appendices are unusually comprehensive, detailed, and clear, which both supports replication and makes clear that great care was taken to anticipate and propose principled solutions to diverse technical challenges that arise in both annotation and modeling for this task.


**Reasons To Reject:**

1. There is a seemingly arbitrary division made between "hyperparameters" (things like learning rate, state size, etc) that are tuned using random search on validation data and "variants" (choices about model architecture and tagging scheme) that are exhaustively evaluated on the test set. These variants also seem to me like hyperparameter choices that will need to be tuned in service of selecting a final working model for this task in any actual application. There are very many "variants" considered here, and as a result I think far too many models were evaluated on the test set (the appendices contain 15 pages of test set results). If this were primarily a modeling paper, I would consider this to be a fishing expedition and a serious flaw. However, this is primarily a resource paper, and this large set of models is framed as a "baseline" showing that the task is tractable to some extent. I still think it would have been better to treat the architecture and tagging scheme as additional hyperparameters and at most only report differences in validation set performance, but I don't think that what was actually done seriously undermines the main contribution.

2. The inter-annotator agreement score for the Latin dataset is low (around 0.4 to 0.6 depending on the measure being tested). The authors are laudably frank about this, and about possible issues with the annotation guidelines that might have contributed to this outcome, and they promise to address this in future annotation efforts. Nonetheless, if rhetoric experts show such low agreement, it raises concerns both about construct validity for this task and about the amount of available headroom for improving on the baseline results for any groups that choose to take this up.

3. I don't understand why a different performance metric was used for early stopping (MBAWO) vs. for the critical evaluations (EPM). Some justification would be helpful.

4. The new resource is in two languages: Latin and Chinese. The Chinese dataset is a light reannotation of an existing dataset, so the bulk of the annotation contribution is for Latin. Although I recognize the historical importance of Latin in the study of rhetoric, I do find it strange that the authors chose to debut this task by concentrating on a language that few living humans use for communication, and I'm having a hard time seeing practical use cases or strong intrinsic motivation for the NLP community to invest in targeting the Latin component of this dataset (targeting Chinese is self-evidently valuable). Is the primary target use case something like writing pedagogy for classics students? In any case, revisions that help clarify the expected practical utility of the Latin portion of the dataset would be welcome.


**Reproducibility:**

5: Could easily reproduce the results.

**Reviewer Confidence:**

3: Pretty sure, but there's a chance I missed something. Although I have a good feel for this area in general, I did not carefully check the paper's details, e.g., the math, experimental design, or novelty.

---

> ### Author Rebuttal · Authors · 2023-08-29
>
> Thank you very much for your review and your meticulous examination of our paper! We're glad that you found it clear, comprehensive, and readily replicable.
>
> We'll start by addressing your concerns about our experimental procedures. As described in our Sections 4.3 and 7.1, we divided the data into four sets: training, validation, optimization, and test. Our hyperparameter search involved eight random trials with each combination of model and tagset (the "variants" which you mention) in which hyperparameters were varied (as described in Appendix F). Each trial saw the training set during training, used the validation set for early stopping, and was tested on the optimization set. Then, the best trial of those eight on the optimization set was deemed the "best". After the hyperparameter search was concluded, only the best trials were evaluated on the test set.
>
> As you mention, we used the MBAWO metric instead of the EPM metric for early stopping. Because EPM is a strict metric, we were concerned that we would overly penalize models for making incremental progress on capturing correct tokens, stopping their training procedures too early. Conversely, being a word-level metric, MBAWO captures that incremental progress more directly. An opposing argument could be made that relying on MBAWO might procure a model that overfits on finding individual words but not whole parallelisms. However, because both metrics innately require parallel structure to achieve a higher score, we can safely wager that MBAWO and EPM scores will correlatively improve with one another. However, because we were being exploratory with our new task, taking advantage of MBAWO's higher granularity seemed like a better fit.
>
> Regarding the hyperparameter search, our rationale behind the division between "variants" and "hyperparameters" was somewhat intuition-based. We felt that the model architecture and the selected tagset would be the most critical values in the hyperparameter search. Thus, we chose to examine them *en masse*. However, we agree that the number of results collected and presented is massive. We wanted to be able to recommend general sets of hyperparameters for future work so that other researchers would have some basis for selecting them in the future. Currently, this recommendation has only been partially achieved, perhaps being hampered by performing too wide of a search. If possible, we would like to expand upon our results section with more statistical and error analyses, thereby providing more justification for our vast results collection.
>
> Turning to your dataset concerns, you are correct that our lower inter-annotator agreement results raise doubts about our task. Inevitably, the only solution for this problem is further annotation studies with revised guidelines. Pertinent future work could involve collecting a larger dataset of parallelism annotations on a living language (*e.g.*, English) both to ascertain our ability to achieve higher performance on this task and to make this task more widely accessible. At the same time, Song *et al.*’s satisfactory annotator agreement score in their sentence-level dataset gives us hope that our token-level Latin dataset can attain a similar reliability. Releasing our data to the research community may aid in its refinement as we continue to improve it ourselves.
>
> Lastly, we wanted to elaborate on potential uses of our dataset. In our paper, we mentioned (and hope to expand upon in the final version) a connection between our work and automatic stylistic analysis (lines 66-68). From a classicist's perspective, parallelism is an important feature of rhetorical Latin prose; its presence or absence forms an important criterion in the comparison of different texts, authors, or even eras of Latin prose writing and speaking. Although classicists closely study stylistic phenomena in individual texts, it is quite time-consuming to cover extensive corpora. (For instance, although our dataset consists of 80 sermons, there are over 800 in Augustine's extant corpus!) Thus, from an NLP perspective, our dataset can assist in a distant reading approach: by training models for rhetorical parallelism, data can be automatically annotated and used toward a diachronic study of rhetorical styles. Features extracted from the performance of RPD could also be used for tasks like authorship attribution, as a proclivity (or lack thereof) for parallelism may help to distinguish one author from another.
>
> These uses, however, have their primary audience as those already studying Latin. We acknowledge your point that it may be difficult to have the wider NLP community invest its efforts in the Latin dataset when their work may concern other living languages. For those whose interest would primarily be on living languages (and who perhaps would endeavor to create data in those languages), our dataset can always assist in showing the generalizability of their proposed methods.
>
> With that said, we want to thank you again for your thorough examination of our work and your careful commentary. If you have any more questions, concerns, or suggestions, please let us know--we'd be glad to hear them!

---

### Official Review · Reviewer_FFhL · 2023-08-03

**Typos Grammar Style And Presentation Improvements:** N/A
**Soundness:** 4

**Excitement:**

4: Strong: This paper deepens the understanding of some phenomenon or lowers the barriers to an existing research direction.

**Missing References:**

N/A

**Paper Topic And Main Contributions:**

This paper focuses on the detection of rhetorical parallelism in Latin and English. Specifically, this paper gives a precise definition of rhetorical parallelism detection (RPD) and proposes two RPD datasets in Latin and Chinese respectively. A bunch of baselines (word2vec, Bi-LSTM, BERT) are tested on proposed datasets. Moderate results demonstrate the challenging of this task. Additionally, this paper proposes a new evaluation metric for this sequence tagging task by calculating the proportion of correct alignments (aligned using Hungarian Algorithm).

**Questions For The Authors:**

Question A: You split your data into training, validation, optimization and test. What is the use of your validation set? I noticed you said in the paper that the optimization set is used for hyper-param tuning but no mention of the use of validation set.

**Reasons To Accept:**

Compared to previous work, the proposed RPD datasets allow token-level parallel structures. The RPD task is re-formulated as a sequence tagging task rather than a classification task, which is novel and interesting. These datasets (together with the new metrics) will be useful for researchers to explore parallelism detection.

The paper is well written and methods are described in great detail.

**Reasons To Reject:**

The agreement between annotators is actually not high, this brings some doubts on the quality of the datasets.

The authors bring in additional tags (e.g., M, E, J), though experiments in Table 5 demonstrate the benefits, the description (line 404 - 431) is not easy to follow and comprehend. The motivation behind thus is not well illustrated.

**Reproducibility:**

4: Could mostly reproduce the results, but there may be some variation because of sample variance or minor variations in their interpretation of the protocol or method.

**Reviewer Confidence:**

3: Pretty sure, but there's a chance I missed something. Although I have a good feel for this area in general, I did not carefully check the paper's details, e.g., the math, experimental design, or novelty.

---

> ### Author Rebuttal · Authors · 2023-08-29
>
> Thank you so much for your review. We're glad that the majority of proposed elements in our paper (*e.g.*, datasets, metrics, models) were presented in a manner that was transparent and straightforward. For our rebuttal, we want to start by addressing some of the points of our work which were less clear--namely, those concerning training and tagging.
>
> First, regarding training, we split our data into four sets: training, validation, optimization, and test (as you mentioned). During training, the validation set was used to determine whether early stopping would occur. After each epoch, we computed MBAWO F1 scores over the validation set; if a model's F1 scores on this set did not improve over 25 epochs, we stopped training early. We recognize that we did not mention the validation set's purpose explicitly in the relevant subsection, and we will make this clearer in the final version.
>
> Second, regarding tagging, the additional tags which we add comprise one of our paper's major novelties. As a result, we certainly want to improve the clarity of the designated section. One critical point to convey is that these tags are intended to divert the model's attention from the majority class of "O" tags. The conditional random field examines all bigram transition probabilities to determine the optimal series of transitions (and thus the optimal series of tags) over the entire input sequence. However, because "O" tags are involved in an overwhelming majority of transitions, a model could easily transition to an "O" tag and discard information about incomplete parallelisms due to the high probability of the transition "O" -> "O".
>
> The "M" tag addresses this issue by decreasing the number of "O" tags overall, replacing them when an "O" tag is between any two related branches. The "E" tag prevents the "I" -> "O" transition by replacing the last "I" tag of any branch with an "E" tag. Lastly, when the "J" tag is paired with the "M" tag, it produces a closed loop of transitions for parallelisms. Each parallelism has an initial branch that starts with a "B" tag; this is followed by 0 or more "I" tags and 0 or more "M" tags; subsequently, there is a linked "B" tag (since each parallelism must have at least two branches) followed by 0 or more "J" tags. Only the linked "B" tag and the "J" tag potentially return to "O" tags, thereby enforcing that at least two branches must be found before the model exits the parallelism.
>
> As a final note, we agree that our inter-annotator agreement ideally would be higher for our dataset. While agreement is better on the word level, achieving a value higher than 0.6 (considered "substantial" agreement by Landis and Koch), we would like to further refine the dataset. We hope that sharing our work with the wider research community can aid in enhancing its quality.
>
> With all that said, we want to thank you again for your time, comments, criticisms, and question. We are happy to answer any further questions or take further suggestions!

---

### Official Review · Reviewer_J4Zn · 2023-08-11

**Soundness:** 4

**Excitement:**

4: Strong: This paper deepens the understanding of some phenomenon or lowers the barriers to an existing research direction.

**Paper Topic And Main Contributions:**

This paper introduces the task of Rhetorical Parallelism Detection. Noting that parallelism is a common rhetorical feature and its presence or absence is a useful indicator for tasks like essay scoring, syntactic parsing, and disfluency detection, the authors annotate a dataset of parallelism in two text corpora, one in Latin and one in Chinese. Detailed definitions of parallelism are provided, and a thorough description of evaluation methods is presented. A reasonable suite of baseline models is tested, showing that good performance is possible but there is room for improvement.

**Reasons To Accept:**

This paper very clearly describes a problem, and then very effectively mobilizes machinery from pre-2020 NLP* to make a well defined machine learning task. The creativity in identifying parallels (kind of meta!) between this task and NER, disfluency detection, coreference resolution is clear, and it means that the metrics and models used are time-tested, well understood, and trustworthy.
I am not an expert in this area but I presume that this dataset will be helpful to digital humanities practitioners in detecting rhetorical devices at a large scale, as well as for automatic essay scoring if it is clear that the essay writers have been encouraged to use parallelism.


* I mean this in a very positive way!

**Reasons To Reject:**

While this paper is very thorough in defining and presenting the RPD task, it is missing some contents that would be interesting and relevant in my opinion. For example, I think it is important for readers to get a 'feel' for a new task, so more examples demonstrating the different types of parallelism (phonology/morphology/syntax/semantics) and their prevalence in the two datasets would be valuable. Since a 40% F1 score is reported, an error analysis would be instructive for future researchers. The authors mention the potential usefulness for downstream tasks; a small demonstration on even a proxy task would help make this claim more convincing.

**Reproducibility:**

5: Could easily reproduce the results.

**Reviewer Confidence:**

4: Quite sure. I tried to check the important points carefully. It's unlikely, though conceivable, that I missed something that should affect my ratings.

---

> ### Author Rebuttal · Authors · 2023-08-29
>
> Thank you so much for your review; we appreciate your kind words about the new task of rhetorical parallelism detection. We are glad that our spin on sequence labeling was well-received.
>
> Concerning the inclusion of more concrete information about the dataset in the paper, we absolutely agree with your suggestions. We had trouble finding space to include it in this draft, but we want to find space in the final version.
>
> In previous drafts of this work, we performed some error analysis and categorized errors into three major types. Although this analysis was derived from an older baseline system, we nevertheless discuss it here to give an example of what one might expect in the final version.
>
> (1) From Sermon 212, Section 1:
> Latin: ... ut iam non aggrauet animam [nec ullam quaerat refectionem]1 , [quia nullam patietur defectionem]1 ...
> English: "... that no longer shall the body burden the soul nor shall it seek any restoration, since it suffers no defect ..."
>
> (2) From Sermon 253, Section 5:
> Latin: <[transcendit nubes]1 et [transcendit sidera]1>1 , <[transcendit angelos]1>1 , <[transcendit omnem creaturam]1>1 <peruenit ad uerbum>1 , per quod facta sunt omnia .
> English: "He transcended clouds and transcended stars, transcended angels, transcended every creature, and he arrived at the Word, through whom all things were made."
>
> (3) From Sermon 186, Section 2:
> Latin: quid enim attenderunt , nisi quia <humana natura potuit in melius commutari>1 , <in deterius autem diuina non potuit>1?
> English: "What, namely, did they consider, except that human nature was able to be transformed into something better, but that divine nature was not able to be transformed into something worse?"
>
> In the examples above, square brackets indicate reference parallelisms (with a numbering indicating the parallelism ID of a given branch relative to the presented passage), whereas angle brackets indicate predicted parallelisms.
>
> In Example (1), we find errors of the first type, in which the model does not predict a parallelism at all (a complete false negative). This was surprising because of the many similarities between the two annotated clauses, as their word order indicates a one-to-one alignment and the grammatical structure is the same. Moreover, the words *ullam* and *nullam* are antonyms ("some/any" and "no/none", respectively), and *refectionem* and *defectionem* are both phonetically alike while playing similar grammatical roles.
>
> In Example (2), we find errors of the second type, in which a model predicts a parallelism in the same location as a reference parallelism, but the predicted parallelism differs from what was annotated. The model fails to bound the individual *transcendit* clauses joined by *et* ("and"), relying too heavily on punctuation for boundaries. It also adds the clause *peruenit ad uerbum*. This clause is grammatically like prior clauses with a similarly-conjugated verb and an accusative noun. However, the accusative noun is an object of a prepositional phrase cued by *ad*; furthermore, it clearly does not include the verb which links the prior clauses: *transcendit*. This error could be described as an "alternate interpretation"--a false positive that is not baseless but has issues.
>
> In Example (3), we find errors of the third type, in which a model predicts a parallelism where none was annotated. However, the model still chooses text that has some merit. The first and second branches contrast *humana natura* with *diuina natura* (the latter *natura* being understood). Both clauses also focus on the same verb, using it both positively (*potuit*) and negatively (*non potuit*). Their substantive adjectives--*in melius* ("into something better") and *in deterius* ("into something worse")--also both parallel one another syntactically while contrasting semantically. In short, there are many similarities between these two clauses; however, the branches' word orders are scrambled, not quite fitting the mold of a parallelism. As a result, it is understandable why our annotator skipped it while our model did not.
>
> We hope that this sample of more data and error analysis clarifies our data better and provides precedent for what one can expect from our paper!
>
> Finally, we agree that additional experiments to show the generalizability of this work would be of interest and, as you say, more convincing of our claims. We have a few ideas for fitting tasks (*e.g.*, Chinese grammatical error detection, Latin syntactic parsing) and, given the time, will look into an addition of this kind.
>
> Thanks again for your time and care in reviewing our paper. We are happy to answer any other questions or hear out further suggestions!

---

### Meta-Review · Area_Chair_m3JP · 2023-09-18

**Recommendation:** 5

**Metareview:**

This paper proposes rhetorical parallelism detection (RPD) as a novel NLP task in which the objective is to identify sets of textual spans in natural language that constitute a parallelism as defined by rhetoreticians. To this end, the authors formalize the task and several plausible evaluation metrics, offer a novel expert-annotated dataset of naturally-occurring parallelisms in Latin and Chinese texts, identify and suggest solutions to important technical challenges (e.g., which tagging scheme to use), and report the performance of a diverse array of baseline models, the results of which show both substantial evidence of learning but also much potential for improvement, suggesting that this is a tractable but challenging task.

In general, the reviewers found RPD and the paper to be very interesting, well-motivated and clearly presented. The scope of the paper is quite wide, including a novel task and formalization, a large expert-annotated resource, and extensive modeling.

One of the main concerns brought up is the low IAA scores. These scores raise doubts about the quality of the dataset, the construct validity for the task, and how much space there is for improvement on a baseline for this task. The authors respond echoing the concerns and elaborating the possible reasons behind this; however, they suggest that releasing the dataset to the community will provide the opportunity for refinement.

Additional questions or need for clarification brought up by reviewers were extensively addressed in the author response. I would expect any camera-ready version to incorporate these details and clarifications.

---

### Decision · Program_Chairs · 2023-10-07

**Decision:**

Accept-Main

**Comment:**

This paper proposes rhetorical parallelism detection (RPD) as a novel NLP task in which the objective is to identify sets of textual spans in natural language that constitute a parallelism as defined by rhetoreticians. To this end, the authors formalize the task and several plausible evaluation metrics, offer a novel expert-annotated dataset of naturally-occurring parallelisms in Latin and Chinese texts, identify and suggest solutions to important technical challenges (e.g., which tagging scheme to use), and report the performance of a diverse array of baseline models, the results of which show both substantial evidence of learning but also much potential for improvement, suggesting that this is a tractable but challenging task.

In general, the reviewers found RPD and the paper to be very interesting, well-motivated and clearly presented. The scope of the paper is quite wide, including a novel task and formalization, a large expert-annotated resource, and extensive modeling.

One of the main concerns brought up is the low IAA scores. These scores raise doubts about the quality of the dataset, the construct validity for the task, and how much space there is for improvement on a baseline for this task. The authors respond echoing the concerns and elaborating the possible reasons behind this; however, they suggest that releasing the dataset to the community will provide the opportunity for refinement.

Additional questions or need for clarification brought up by reviewers were extensively addressed in the author response. I would expect any camera-ready version to incorporate these details and clarifications.